# INVERSE FLOW AND CONSISTENCY MODELS

## ABSTRACT

Inverse generation problems, such as denoising without ground truth observations, is a critical challenge in many scientific inquiries and real-world applications. While recent advances in generative models like diffusion models, conditional flow matching, and consistency models achieved impressive results by casting generation as denoising problems, they cannot be directly used for inverse generation without access to clean data. Here we introduce Inverse Flow (IF), a novel framework that enables using these generative models for inverse generation problems including denoising without ground truth. Inverse Flow can be flexibly applied to nearly any continuous noise distribution and allows complex dependencies. We propose two algorithms for learning Inverse Flows, Inverse Flow Matching (IFM) and Inverse Consistency Model (ICM). Notably, to derive the computationally efficient, simulation-free inverse consistency model objective, we generalized consistency training to any forward diffusion processes or conditional flows, which have applications beyond denoising. We demonstrate the effectiveness of IF on synthetic and real datasets, outperforming prior approaches while enabling noise distributions that previous methods cannot support. Finally, we showcase applications of our techniques to fluorescence microscopy and single-cell genomics data, highlighting IF's utility in scientific problems. This work opens up the use of powerful generative models for inversion generation problems.

## 1 INTRODUCTION

Recent advances in generative modeling such as diffusion models (Sohl-Dickstein et al., 2015; Ho et al., 2020; Song & Ermon, 2020; Song et al., 2021; 2022), conditional flow matching models (Lipman et al., 2023; Tong et al., 2024), and consistency models (Song et al., 2023; Song & Dhariwal, 2023) have achieved great success by learning a mapping from a simple prior distribution to the data distribution through an Ordinary Differential Equation (ODE) or Stochastic Differential Equation (SDE). We refer to them models as continuous-time generative models. These models typically involve defining a forward process, which transforms the data distribution to the prior distribution over time, and generation is achieved through learning a reverse process that can gradually transform the prior distribution to the data distribution (Figure 1).

Despite that those generative models are powerful tools for modeling the data distribution, they are not suitable for the *inverse generation problems* when the data distribution is not observed and only data transformed by a forward process is given, which is typically true for noisy real-world data measurements. Mapping from noisy data to the latent ground truth is especially important in various scientific applications when pushing the limit of measurement capabilities. This limitation necessitates the exploration of novel methodologies that can bridge the gap between generative modeling and effective denoising in the absence of clean data.

Here we propose a new approach called **Inverse Flow (IF)**, that learns a mapping from the observed noisy data distribution to the unobserved, ground truth data distribution (Figure 1), **inverting** the data requirement of generative models. An ODE or SDE is specified to reflect knowledge about the noise distribution. We further devised a pair of algorithms, **Inverse Flow Matching (IFM)** and **Inverse Consistency Model (ICM)** for learning inverse flows. Specifically, ICM involves a computationally efficient simulation-free objective that does not involve any ODE solver.

A main contribution of our approach is generalizing continuous-time generative models to inverse generation problems such as denoising without ground truth. In addition, in order to develop ICM,

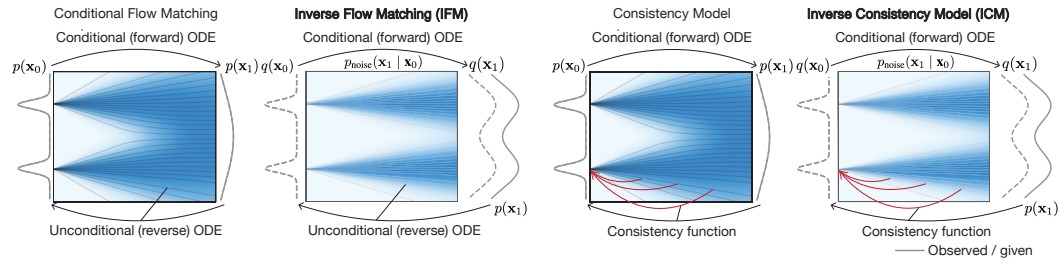

Figure 1: Inverse flow enables adapting the family continuous-time generative models for solving inverse generation problems. Inverse flow algorithms (inverse flow matching and inverse consistency model) are built upon conditional flow matching and consistency models respectively.

we generalized the consistency training objective for consistency models to any forward diffusion process or conditional flow. This broadens the scope of consistency model applications and has implications beyond denoising.

Compared to prior approaches for denoising without ground truth, IF offers the most flexibility in noise distribution, allowing almost any continuous noise distributions including those with complex dependency and transformations. IF can be seamlessly integrated with generative modeling to generate samples from the ground truth rather than the observed noisy distribution. More generally, IF models the past states of a (stochastic) dynamical system before the observed time points using the knowledge of its dynamics, which can have applications beyond denoising.

## 2 BACKGROUND

### 2.1 CONTINUOUS-TIME GENERATIVE MODELS

Our proposed inverse flow framework is built upon continuous-time generative models such as diffusion models, conditional flow matching, and consistency models. Here we present a unified view of these methods that will help connect inverse flow with this entire family of models (Section 3).

These generative modeling methods are connected by their equivalence to continuous normalizing flow or neural ODE (Chen et al., 2019). They can all be considered as explicitly or implicitly learning the ODE that transforms between the prior distribution $p(\mathbf{x}_1)$ and the data distribution $p(\mathbf{x}_0)$

$$\mathrm{d}\mathbf{x} = \mathbf{u}_t(\mathbf{x})\mathrm{d}t. \tag{1}$$

in which $\mathbf{u}_t(\mathbf{x})$ represents the vector field of the ODE. We use the convention that $t = 0$ corresponds to the data distribution and $t = 1$ corresponds to the prior distribution. Generation is realized by reversing this ODE, which makes this family of methods a natural candidate for extension toward denoising problems.

Continuous-time generative models typically involve defining a conditional ODE or SDE that determines the $p(\mathbf{x}_t|\mathbf{x}_0)$ that transforms the data distribution to the prior distribution. Training these models involves learning the unconditional ODE (Eq. 1) based on $\mathbf{x}_0$ and the conditional ODE or SDE (Lipman et al., 2023; Tong et al., 2024; Song et al., 2021) (Figure 1). The unconditional ODE can be used for generation from noise to data.

### 2.1.1 CONDITIONAL FLOW MATCHING

Conditional flow matching defines the transformation from data to prior distribution via a conditional ODE vector field $\mathbf{u}_t(\mathbf{x} \mid \mathbf{x}_0)$. The unconditional ODE vector field $\mathbf{v}_t^\theta(\mathbf{x})$ is learned by minimizing the objective (Lipman et al., 2023; Tong et al., 2024; Albergo & Vanden-Eijnden, 2023):

$$\left\| \mathbf{v}_t^\theta(\mathbf{x}_t) - \mathbf{u}_t\left(\mathbf{x}_t \mid \mathbf{x}_0\right) \right\|. \tag{2}$$

where $\mathbf{x}_0$ is sampled from the data distribution, and $\mathbf{x}_t$ is sampled from the conditional distribution $p(\mathbf{x}_t \mid \mathbf{x}_0)$ given by the conditional ODE.

The conditional ODE vector field $\mathbf{u}_t(\mathbf{x} \mid \mathbf{x}_0)$ can also be stochastically approximated through sampling from both prior distribution and data distribution and using the conditional vector field $\mathbf{u}_t(\mathbf{x} \mid \mathbf{x}_0, \mathbf{x}_1)$ as the training target (Lipman et al., 2023; Tong et al., 2024):

$$\left\| \mathbf{v}_t^\theta(\mathbf{x}_t) - \mathbf{u}_t\left(\mathbf{x}_t \mid \mathbf{x}_0, \mathbf{x}_1\right) \right\|. \tag{3}$$

This formulation has the benefit that $\mathbf{u}_t(\mathbf{x} \mid \mathbf{x}_0, \mathbf{x}_1)$ can be easily chosen as any interpolation between $\mathbf{x}_0$ and $\mathbf{x}_1$, because this interpolation does not affect the probability density at time 0 or 1 (Lipman et al., 2023; Tong et al., 2024; Albergo & Vanden-Eijnden, 2023; Albergo et al., 2023). For example, a linear interpolation corresponds to $\mathbf{x}_t = \mathbf{x}_0 + t(\mathbf{x}_1 - \mathbf{x}_0)$ (Lipman et al., 2023; Tong et al., 2024; Liu et al., 2022). Sampling is realized by simulating the unconditional ODE with learned vector field $\mathbf{v}_t^\theta(\mathbf{x})$ in the reverse direction.

### 2.1.2 CONSISTENCY MODELS

In contrast, consistency models (Song et al., 2023; Song & Dhariwal, 2023) learn consistency functions that can directly map a sample from the prior distribution to data distribution, equivalent to simulating the unconditional ODE in the reverse direction:

$$\mathbf{c}(\mathbf{x}_t, t) = \mathrm{ODE}_{t \to 0}^{\mathbf{u}}(\mathbf{x}_t)$$

where $\mathbf{x}_t$ denotes $\mathbf{x}$ at time $t$, and we use $\mathrm{ODE}_{t \to 0}^{\mathbf{u}}(\mathbf{x}_t)$ to denote simulating the ODE with vector field $\mathbf{u}_t(\mathbf{x})$ from time $t$ to time 0 starting from $\mathbf{x}_t$. The consistency function is trained by minimizing the consistency loss (Song et al., 2023), which measures the difference between consistency function evaluations at two adjacent time points

$$\mathcal{L}_{\mathrm{CM}}(\theta) = \mathbb{E}_{i, \mathbf{x}_{t_i}, \mathbf{x}_{t_{i+1}}} \left[ \left\| \mathbf{c}_\theta(\mathbf{x}_{t_{i+1}}, t_{i+1}) - \mathrm{stopgrad}\left(\mathbf{c}_\theta(\mathbf{x}_{t_i}, t_i)\right) \right\| \right] \tag{4}$$

with the boundary condition $\mathbf{c}(\mathbf{x}, 0) = \mathbf{x}$. Stopgrad indicates that the term within the operator does not get optimized.

There are two approaches to training consistency models: one is distillation, and the other is training from scratch. In the consistency distillation objective, a pretrained diffusion model is used to obtain the unconditional ODE vector field $\mathbf{u}_t$, and $\mathbf{x}_{t_{i+1}}$ and $\mathbf{x}_{t_i}$ differs by one ODE step

$$\mathbf{x}_{t_{i+1}} \sim p(\mathbf{x}_{t_{i+1}} \mid \mathbf{x}_0), \quad \mathbf{x}_{t_{i+1}} - \mathbf{x}_{t_i} = \mathbf{u}_{t_{i+1}}(\mathbf{x}_{t_{i+1}})(t_{i+1} - t_i) \tag{5}$$

If the consistency model is trained from scratch, the consistency training objective samples $\mathbf{x}_{t_{i+1}}$ and $\mathbf{x}_{t_i}$ in a coupled manner from the forward diffusion process (Karras et al., 2022)

$$\mathbf{x}_{t_{i+1}} = \mathbf{x}_0 + \mathbf{z} t_{i+1}, \quad \mathbf{x}_{t_i} = \mathbf{x}_0 + \mathbf{z} t_i, \quad \mathbf{z} \sim \mathcal{N}(0, \sigma^2 \mathbf{I}) \tag{6}$$

where $\sigma$ controls the maximum noise level at $t = 1$. Consistency models have the advantage of fast generation speed as they can generate samples without solving any ODE or SDE.

### 2.1.3 DIFFUSION MODELS

In diffusion models, the transformation from data to prior distribution is defined by a forward diffusion process (conditional SDE). The diffusion model training learns the score function which determines the unconditional ODE, also known as the probability flow ODE (Song et al., 2021).

**Denoising applications of diffusion models** Diffusion models are inherently connected to denoising problems as the generation process is essentially a denoising process. However, existing denoising methods using diffusion models require training on ground truth data (Yue et al., 2023; Xie et al., 2023b), which is not available in inverse generation problems.

**Ambient diffusion and GSURE-diffusion** Ambient Diffusion (Daras et al., 2023) and GSURE-diffusion (Kawar et al., 2024) address a related problem of learning the distribution of clean data by training on only linearly corrupted (linear transformation followed by additive Gaussian noise) data. Although those methods are designed for generation, they can be applied to denoising. Ambient Diffusion Posterior Sampling (Aali et al., 2024), further allowed using models trained with ambient

diffusion on corrupted data to perform posterior sampling-based denoising for a different forward process (e.g., blurring). Consistent Diffusion Meets Tweedie (Daras et al., 2024) improves Ambient Diffusion by allowing exact sampling from clean data distribution using consistency loss with a double application of Tweedie's formula. Rozet et al. (2024) explored the potential of expectation maximization in training diffusion models on corrupted data. However, all these methods are restricted to training on linearly corrupted data, which still limit their applications when the available data is affected by other types of noises.

## 2.2 Denoising without ground truth

Denoising without access to ground truth data requires assumptions about the noise or the signal. Most contemporary approaches are based on assumptions about the noise, as the noise distribution is generally much simpler and better understood. Because prior methods have been comprehensively reviewed (Kim & Ye, 2021; Batson & Royer, 2019; Lehtinen et al., 2018; Xie et al., 2020; Soltanayev & Chun, 2018; Metzler et al., 2020), and our approach is not directly built upon these approaches, we only present a brief overview and refer the readers to Appendix A.3 referenced literature for more detailed discussion. None of these approaches are generally applicable to any noise types.

## 3 Inverse Flow and Consistency Models

In continuous-time generative models, usually the data $\mathbf{x}_0$ from the distribution of interest is given. In contrast, in inverse generation problems, only the transformed data $\mathbf{x}_1$ and the conditional distribution $p(\mathbf{x}_1|\mathbf{x}_0)$ are given, whereas $\mathbf{x}_0$ are unobserved. For example, $\mathbf{x}_1$ are the noisy observations and $p(\mathbf{x}_1|\mathbf{x}_0)$ is the conditional noise distribution. We define the *Inverse Flow* (IF) problem as finding a mapping from $\mathbf{x}_1$ to $\mathbf{x}_0$ which allows not only recovering the unobserved data distribution $p(\mathbf{x}_0)$ but also providing an estimate of $\mathbf{x}_0$ from $\mathbf{x}_1$ (Figure 1).

For denoising without ground truth applications, the inverse flow framework requires only the noisy data $\mathbf{x}_1$ and the ability to sample from the noise distribution $p(\mathbf{x}_1|\mathbf{x}_0)$. This is thus applicable to any continuous noise and allows complex dependencies on the noise distribution, including noise that can only be sampled through a diffusion process.

### 3.1 Inverse Flow Matching

To solve the inverse flow problem, we first consider learning a mapping from $\mathbf{x}_1$ to $\mathbf{x}_0$ through an ODE with vector field $\mathbf{v}_t^\theta(\mathbf{x})$. We propose to learn $\mathbf{v}_t^\theta(\mathbf{x})$ with the inverse flow matching (IFM) objective

$$\mathcal{L}_{\text{IFM}}(\theta) = \mathbb{E}_{t,p(\mathbf{x}_1),p\left(\mathbf{x}_t|\mathbf{x}_0=\text{ODE}_{1\to0}^{\mathbf{v}^\theta}(\mathbf{x}_1)\right)} \left\| \mathbf{v}_t^\theta(\mathbf{x}_t) - \mathbf{u}_t\left(\mathbf{x}_t \mid \text{ODE}_{1\to0}^{\mathbf{v}^\theta}(\mathbf{x}_1)\right) \right\| \qquad (7)$$

This objective differs from conditional flow matching (Eq. 2) in two key aspects: using only transformed data $\mathbf{x}_1$ rather than unobserved data $\mathbf{x}_0$, and choosing the conditional ODE based on the conditional distribution $p(\mathbf{x}_1|\mathbf{x}_0)$. Specifically,

1. Sampling from the data distribution $p(\mathbf{x}_0)$ is replaced with sampling from $p(\mathbf{x}_1)$ and simulating the unconditional ODE backward in time based on the vector field $\mathbf{v}$, denoted as $\text{ODE}_{t\to0}^{\mathbf{v}^\theta}(\mathbf{x}_1)$. We refer to this distribution as the recovered data distribution $q(\mathbf{x}_0)$.

2. The conditional ODE vector field $\mathbf{u}_t(\mathbf{x} \mid \mathbf{x}_0)$ is chosen to match the given conditional distribution $p(\mathbf{x}_1|\mathbf{x}_0)$ at time 1.

For easier and more flexible application of IFM, similar to conditional flow matching (Eq. 3), an alternative form of the conditional ODE $\mathbf{u}_t(\mathbf{x} \mid \mathbf{x}_0, \mathbf{x}_1')$ can be used instead of $\mathbf{u}_t(\mathbf{x} \mid \mathbf{x}_0)$. Since $\mathbf{x}_1'$ is sampled from the noise distribution $p(\mathbf{x}_1|\mathbf{x}_0)$, the above condition is automatically satisfied. The conditional ODE vector field can be easily chosen as any smooth interpolation between $\mathbf{x}_0$ and $\mathbf{x}_1'$, such as $\mathbf{u}_t(\mathbf{x} \mid \mathbf{x}_0, \mathbf{x}_1') = \mathbf{x}_1' - \mathbf{x}_0$. We detailed the inverse flow matching training in Algorithm 1 with the alternative form in Appendix A.1.

| **Algorithm 1** IFM Training | **Algorithm 2** ICM Training |
|---|---|
| 1: **Input:** dataset $\mathcal{D}$, initial model parameter $\theta$, and learning rate $\eta$ | 1: **Input:** dataset $\mathcal{D}$, initial model parameter $\theta$, learning rate $\eta$, and sequence of time points $0 = t_1 < t_2 < \cdots < t_N = 1$ |
| 2: **repeat** | 2: **repeat** |
| 3:    Sample $\mathbf{x}_1 \sim \mathcal{D}$ and $t \sim \mathcal{U}[0,1]$ | 3:    Sample $\mathbf{x}_1 \sim \mathcal{D}$ and $i \sim \mathcal{U}[1, N-1]$ |
| 4:    $\mathbf{x}_0 \leftarrow \text{stopgrad}\left(\text{ODE}_{1\to 0}^{\mathbf{v}^\theta}(\mathbf{x}_1)\right)$ | 4:    $\mathbf{x}_0 \leftarrow \text{stopgrad}\left(\mathbf{c}_\theta(\mathbf{x}_1, 1)\right)$ |
| 5:    Sample $\mathbf{x}_t \sim p(\mathbf{x}_t \mid \mathbf{x}_0)$ | 5:    Sample $\mathbf{x}_{t_{i+1}} \sim p(\mathbf{x}_{t_{i+1}} \mid \mathbf{x}_0)$ |
| 6:    $\mathcal{L}(\theta) \leftarrow \left\| \mathbf{v}_t^\theta(\mathbf{x}_t) - \mathbf{u}_t(\mathbf{x}_t \mid \mathbf{x}_0) \right\|$ | 6:    $\mathbf{x}_{t_i} \leftarrow \mathbf{x}_{t_{i+1}} - \mathbf{u}_{t_{i+1}}(\mathbf{x}_{t_{i+1}} \mid \mathbf{x}_0)(t_{i+1} - t_i)$ |
| 7:    $\theta \leftarrow \theta - \eta \nabla_\theta \mathcal{L}(\theta)$ | 7:    $\mathcal{L}(\theta) \leftarrow$ $\left\| \mathbf{c}_\theta(\mathbf{x}_{t_{i+1}}, t_{i+1}) - \text{stopgrad}\left(\mathbf{c}_\theta(\mathbf{x}_{t_i}, t_i)\right) \right\|$ |
| 8: **until** convergence | 8:    $\theta \leftarrow \theta - \eta \nabla_\theta \mathcal{L}(\theta)$ |
| | 9: **until** convergence |

Next, we discuss the theoretical justifications of the IFM objective and the interpretation of the learned model. We show below that when the loss converges, the recovered data distribution $q(\mathbf{x}_0)$ matches the ground truth distribution $p(\mathbf{x}_0)$. The proof is provided in Appendix A.2.1.

**Theorem 1** *Assume that the noise distribution $p(\mathbf{x}_1 \mid \mathbf{x}_0)$ satisfies the condition that, for any noisy data distribution $p(\mathbf{x}_1)$ there exists only one probability distribution $p(\mathbf{x}_0)$ that satisfies $p(\mathbf{x}_1) = \int p(\mathbf{x}_1 \mid \mathbf{x}_0)p(\mathbf{x}_0)\mathrm{d}\mathbf{x}_0$ , then under the condition that $\mathcal{L}_{\text{IFM}}$ is minimized, we have the recovered data distribution $q(\mathbf{x}_0) = p(\mathbf{x}_0)$.*

Moreover, we show that with IFM the learned ODE trajectory from $\mathbf{x}_1$ to $\mathbf{x}_0$ can be intuitively interpreted as always pointing toward the direction of the estimated $\mathbf{x}_0$. More formally, the learned unconditional ODE vector field can be interpreted as an expectation of the conditional ODE vector field.

**Lemma 1** *Given a conditional ODE vector field $\mathbf{u}_t(\mathbf{x} \mid \mathbf{x}_0, \mathbf{x}_1)$ that generates a conditional probability path $p(\mathbf{x}_t \mid \mathbf{x}_0, \mathbf{x}_1)$, the unconditional probability path $p(\mathbf{x}_t)$ can be generated by the unconditional ODE vector field $\mathbf{u}_t(\mathbf{x})$, which is defined as*

$$\mathbf{u}_t(\mathbf{x}) = \mathbb{E}_{p(\mathbf{x}_0, \mathbf{x}_1 \mid \mathbf{x})}\left[\mathbf{u}_t(\mathbf{x} \mid \mathbf{x}_0, \mathbf{x}_1)\right] \tag{8}$$

The proof is provided in Appendix A.2.1. Specifically, with the choice of $\mathbf{u}_t(\mathbf{x} \mid \mathbf{x}_0, \mathbf{x}_1) = \mathbf{x}_1 - \mathbf{x}_0$, Eq. 8 has an intuitively interpretable form

$$\mathbf{u}_t(\mathbf{x}) = \mathbb{E}_{p(\mathbf{x}_0 \mid \mathbf{x})}\left[\frac{\mathbf{x} - \mathbf{x}_0}{t}\right] \tag{9}$$

which means that the unconditional ODE vector field at any time $t$ points straight toward the expected ground truth $\mathbf{x}_0$.

## 3.2 SIMULATION-FREE INVERSE FLOW WITH INVERSE CONSISTENCY MODEL

IFM can be computationally expensive during training and inference because it requires solving ODE in each update. We address this limitation by introducing inverse consistency model (ICM), which learns a consistency function to directly solve the inverse flow without involving an ODE solver.

However, the original consistency training formulation is specific to one type of diffusion process (Karras et al., 2022), which is *only applicable to independent Gaussian noise distribution* for inverse generation application. Thus, to derive inverse consistency model that is applicable to any transformation, we first generalize consistency training so that it can be applied to arbitrary transformations and thus flexible to model almost any noise distribution.

### 3.2.1 GENERALIZED CONSISTENCY TRAINING

To recall from Section 2.1.2, consistency distillation is only applicable to distilling a pretrained diffusion or conditional flow matching model. The consistency training objective allows training

consistency models from scratch but only for a specific forward diffusion process, which limits its flexibility in applying to any inverse generation problem.

**Generalized Consistency Training** $\longleftrightarrow$ Consistency Distillation

Conditional Flow Matching $\longleftrightarrow$ Flow Matching

Here we introduce generalized consistency training (GCT), which extends consistency training to any conditional ODE or forward diffusion process (through the corresponding conditional ODE). Intuitively, generalized consistency training modified consistency distillation in the same manner as how conditional flow matching modified the flow matching objective. It differs from consistency distillation (Eq. 4 and Eq. 5) in that it only requires the conditional ODE vector field $\mathbf{u}_t(\mathbf{x} \mid \mathbf{x}_0)$ which is user-specified rather than the unconditional ODE vector field $\mathbf{u}_t(\mathbf{x})$ which has to be learned via a pretrained diffusion or conditional flow matching model.

$$\mathcal{L}_{\mathrm{GCT}}(\theta) = \mathbb{E}_{i,p(\mathbf{x}_0),p(\mathbf{x}_{t_{i+1}}|\mathbf{x}_0)} \left\| \left(\mathbf{c}_\theta(\mathbf{x}_{t_{i+1}}, t_{i+1}) - \mathrm{stopgrad}\left(\mathbf{c}_\theta(\mathbf{x}_{t_i}, t_i)\right)\right)\right\|,$$
$$\mathbf{x}_{t_{i+1}} - \mathbf{x}_{t_i} = \mathbf{u}_{t_{i+1}}(\mathbf{x}_{t_{i+1}} \mid \mathbf{x}_0)(t_{i+1} - t_i) \tag{10}$$

Or we can use the alternative formulation where the conditional flow is defined by $\mathbf{u}_{t_{i+1}}(\mathbf{x} \mid \mathbf{x}_0, \mathbf{x}_1)$ with details in Appendix A.1.

We proved that the generalized consistency training (GCT) objective is equivalent to the consistency distillation (CD) objective (Eq. 4, Eq. 5). The proof is provided in Appendix A.2.2.

**Theorem 2** *Assuming the consistency function $\mathbf{c}_\theta$ is twice differentiable, up to a constant independent of $\theta$, $\mathcal{L}_{\mathrm{GCT}}$ and $\mathcal{L}_{\mathrm{CD}}$ are equal.*

### 3.2.2 INVERSE CONSISTENCY MODELS

With generalized consistency training, we can now provide the inverse consistency model (ICM) (Figure 1, Algorithm 2):

$$\mathcal{L}_{\mathrm{ICM}}(\theta) = \mathbb{E}_{i,p(\mathbf{x}_1),p\left(\mathbf{x}_{t_{i+1}}|\mathbf{x}_0 = \mathbf{c}_\theta(\mathbf{x}_1,1)\right)} \left\| \left(\mathbf{c}_\theta(\mathbf{x}_{t_{i+1}}, t_{i+1}) - \mathrm{stopgrad}\left(\mathbf{c}_\theta(\mathbf{x}_{t_i}, t_i)\right)\right)\right\|,$$
$$\mathbf{x}_{t_{i+1}} - \mathbf{x}_{t_i} = \mathbf{u}_{t_{i+1}}(\mathbf{x}_{t_{i+1}} \mid \mathbf{x}_0)(t_{i+1} - t_i) \tag{11}$$

which is the consistency model counterpart of IFM (Eq. 7). Similar to IFM, a convenient alternative form is provided in Appendix A.1.

Since learning a consistency model is equivalent to learning a conditional flow matching model, ICM is equivalent to IFM following directly from our Theorem 2 and Theorem 1 from Song et al. (2023), but it is much more computationally efficient as it is a simulation-free objective.

## 4 EXPERIMENTS

We first demonstrated the performance and properties of IFM and ICM on synthetic inverse generation datasets, which include a deterministic problem of inverting Naiver-Stokes simulation and a stochastic problem of denoising a synthetic noise dataset 8-gaussians. Next, we demonstrated that our method outperforms prior methods (Mäkinen et al., 2020; Krull et al., 2019; Batson & Royer, 2019) with the same neural network architecture on a semi-synthetic dataset of natural images with three synthetic noise types, and a real-world dataset of fluorescence microscopy images. Finally, we demonstrated that our method can be applied to denoise single-cell genomics data.

### 4.1 SYNTHETIC DATASETS

To test the capability of inverse flow in inverting complex transformations, we first attempted the deterministic inverse generation problem of inverting the transformation by Navier-Stokes fluid

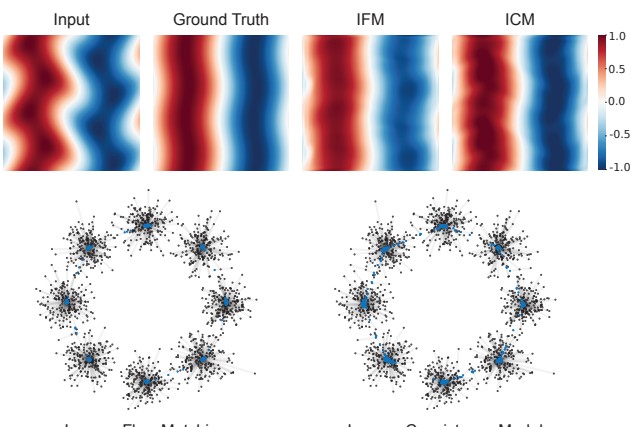

Figure 2: Demonstration of inverse flow algorithms on synthetic datasets. Top panel shows an application to inverting Navier-Stokes fluid dynamics simulation color indicating horizontal velocity. Bottom panel shows a denoising application on 8-gaussians dataset with input (black) and denoised data (blue) connected with lines.

Table 1: Quantitative benchmark of denoising performances in multiple datasets for various noise distributions measured by Peak signal-to-noise ratio (PSNR) in dB

| Noise type | | Input | Supervised | BM3D | Noise2Void | Noise2Self | **Ours (ICM)** |
|---|---|---|---|---|---|---|---|
| | BSDS500 | 20.17 | 28.00 | 27.49 | 26.54 | 27.79 | **28.16** |
| Gaussian | Kodak | 20.18 | 28.91 | 28.54 | 27.55 | 28.72 | **29.08** |
| | Set12 | 20.16 | 28.99 | 28.95 | 27.79 | 28.78 | **29.19** |
| | BSDS500 | 20.17 | 27.10 | 24.48 | 26.32 | 21.03 | **27.64** |
| Correlated | Kodak | 20.17 | 27.97 | 25.03 | 27.39 | 21.56 | **28.53** |
| | Set12 | 20.18 | 27.88 | 25.21 | 27.43 | 21.58 | **28.46** |
| | BSDS500 | 14.90 | **24.34** | 20.32 | 23.56 | 22.60 | 24.28 |
| SDE (Jacobi process) | Kodak | 14.76 | **25.34** | 20.42 | 23.99 | 23.70 | 25.07 |
| | Set12 | 14.80 | **25.01** | 20.51 | 24.43 | 23.26 | 24.74 |

dynamics simulation[1]. We aim to recover the earlier state of the system without providing them for training (Figure 2). Navier-Stokes equations describe the motion of fluids by modeling the relationship between fluid velocity, pressure, viscosity, and external forces. These equations are fundamental in fluid dynamics and remain mathematically challenging, particularly in understanding turbulent flows. The details of the simulation are described in Appendix A.4.2.

To test inverse flow algorithms on a denoising inverse generation problem, we generated a synthetic 8-gaussians dataset (Appendix A.4.2 for details). Both IFM and ICM are capable of noise removal (Figure 2). ICM achieved a similar denoising performance as IFM, even though it is much more computationally efficient due to the iterative evaluation of ODE (NFE=10) by IFM.

## 4.2 SEMI-SYNTHETIC DATASETS

We evaluated the proposed method on images in the benchmark dataset BSDS500 (Arbeláez et al., 2011), Kodak, and Set12 (Zhang et al., 2017). To test the model's capability to deal with various types of conditional noise distribution, we generated synthetic noisy images for three different types of noise, including correlated noise and adding noise through a diffusion process without a closed-form transition density function (Appendix A.4.3 for details). All models were trained using the BSDS500 training set and evaluated on the BSDS500 test set, Kodak, and Set12. We show additional qualitative results in Appendix A.6.

1. Gaussian noise: we added independent Gaussian noise with fixed variance.

---

[1]Inverse flow algorithms can be applied to deterministic transformations from $\mathbf{x}_0$ to $\mathbf{x}_1$ by using a matching conditional ODE, even though the general forms consider stochastic transforms described by $p(\mathbf{x}_1 \mid \mathbf{x}_0)$.

2. Correlated noise: we employed convolution kernels to generate correlated Gaussian noise following the method in Mäkinen et al. (2020)

$$\eta = \nu \circledast g \tag{12}$$

where $\nu \sim \mathcal{N}(0, \sigma^2 I)$ and $g$ is a convolution kernel.

3. Jacobi process: we transformed the data with Jacobi process (Wright-Fisher diffusion), as an example of SDE-based transform without closed-form conditional distribution

$$d\mathbf{x} = \frac{s}{2}[a(1 - \mathbf{x}) - b\mathbf{x}]dt + \sqrt{s\mathbf{x}(1 - \mathbf{x})}d\mathbf{w}. \tag{13}$$

We generated corresponding noise data by simulating the Jacobi process with $s = 1$ and $a = b = 1$. Notably, the conditional noise distribution generated by the Jacobi process does not generally has an expectation that equals the ground truth (i.e. non-centered noise), which violates the assumptions of Noise2X methods.

Our approach outperformed alternative unsupervised methods in all three noise types, especially in correlated noise and Jacobi process (Appendix A.6, Table 4.2). This can be attributed to the fact that both Noise2X methods assumes independence of noise across different feature dimensions as well as centered-noise which were violated in corrleated noise and Jacobi process respectively.

Moreover, Our approach outperformed the supervised method on both Gaussian noise and correlated noise. Further analysis revealed that the supervised method encountered overfitting during the training process, which led to suboptimal performance. In contrast, our method did not exhibit such issues, highlighting the superiority of our approach.

In addition, in Appendix A.5, we conducted a series of experiments that demonstrate the reliability of our method under different intensities and types of noise. Furthermore, our method yielded satisfactory results even when there is a bias in the estimation of noise intensity. It also achieved excellent performance on RGB images and small sample-size datasets.

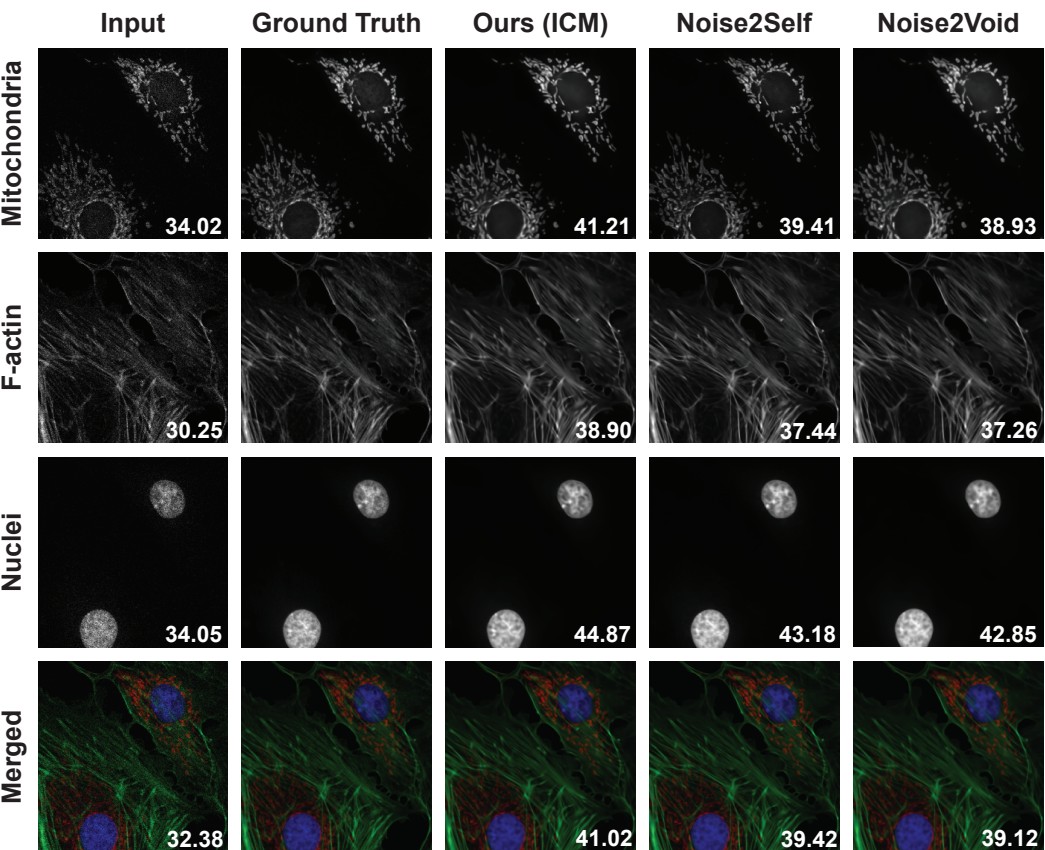

Figure 3: Denoising results for fluorescence microscopy images with PSNR labelled.

### 4.3 REAL-WORLD DATASETS

#### 4.3.1 FLUORESCENCE MICROSCOPY DATA (FMD)

Fluorescence microscopy is an important scientific application of denoising without ground truth. Experimental constraints such as phototoxicity and frame rates often limit the capability to obtain clean data. We denoised confocal microscopy images from Fluorescence Microscopy Denoising (FMD) dataset (Zhang et al., 2019). We first fitted a signal-dependent Poisson-Gaussian noise model adopted from Liu et al. (2013) for separate channels of each noisy microscopic images (Appendix A.4.4 for details). Then denoising flow models were trained with the conditional ODE specified to be consistent with fitted noise model. Our method outperforms Noise2Self and Noise2Void, achieving superior denoising performance for mitochondria, F-actin, and nuclei in the microscopic images of BPAE cells.

#### 4.3.2 APPLICATION TO DENOISE SINGLE-CELL GENOMICS DATA

In recent years, the development of single-cell sequencing technologies has enabled researchers to obtain more fine-grained information on tissues and organs at the resolution of single cells. However,

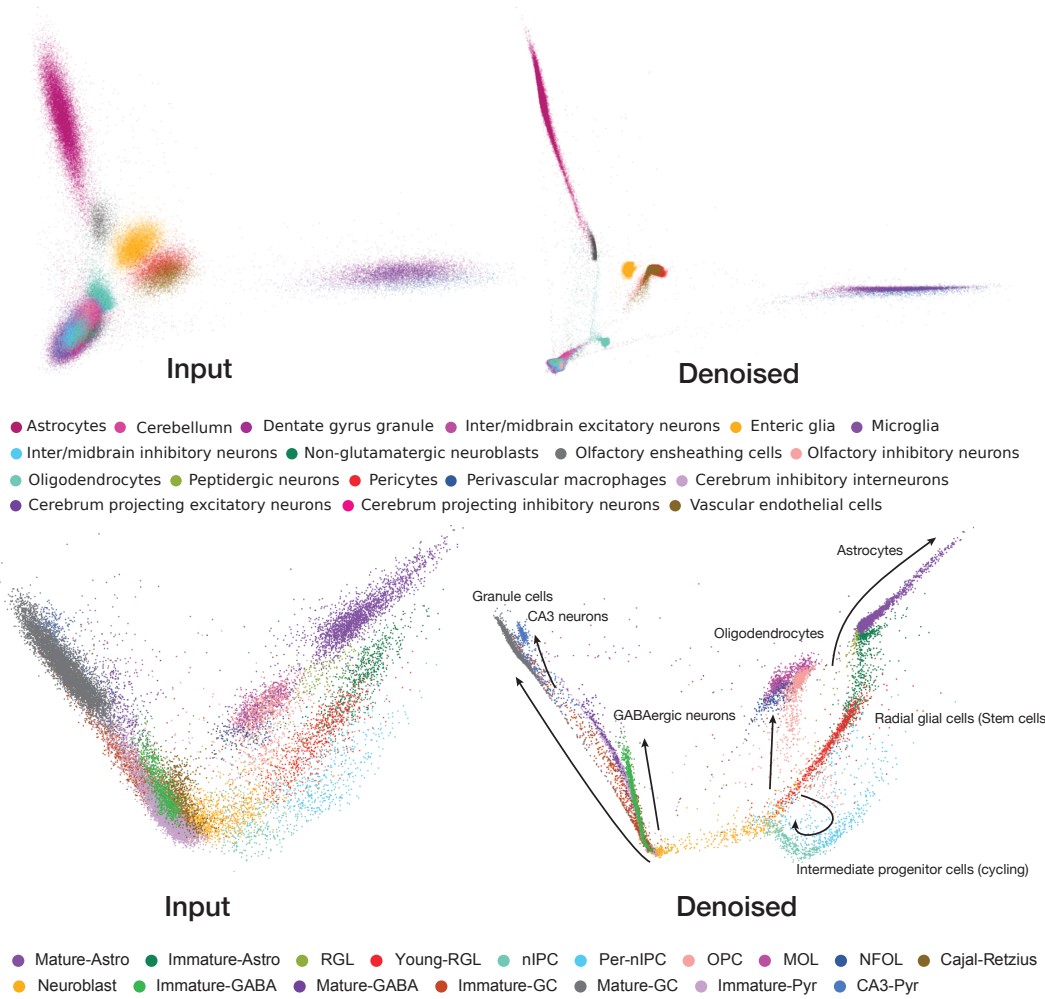

Figure 4: Denoising single-cell RNA-seq data with ICM improves resolution for cell types and developmental trajectories. The top two principal components are visualized. Top panel: results for Zeisel et al. (2018). Bottom panel: results for Hochgerner et al. (2018b), Astro: astrocytes, RGL: radial glial cells, IPC: intermediate progenitor cells, OPC: oligodendrocyte precursor cells, MOL: mature oligodendrocytes; NFOL: newly formed oligodendrocytes, GABA: GABAergic neurons, GC: granule cells, Pyr: pyramidal neurons.

the low amount of sample materials per-cell introduces considerable noise in single-cell genomics data. These noises may obscure real biological signals, thereby affecting subsequent analyses.

Applying ICM to an adult mouse brain single-cell RNA-seq dataset (Zeisel et al., 2018) and a mouse brain development single-cell RNA-seq dataset (Hochgerner et al., 2018b) (Figure 4, Appendix A.4.5 for details), we observed that the denoised data better reflects the cell types and developmental trajectories. We compared the original and denoised data by the accuracy of predicting the cell type identity of each cell based on its nearest neighbor in the top two principal components. Our methods improved the accuracy of the adult mouse brain dataset from $0.513 \pm 0.003$ to $0.571 \pm 0.003$, and the mouse brain development dataset from $0.647 \pm 0.006$ to $0.736 \pm 0.006$.

## 5 LIMITATION AND CONCLUSION

We introduce Inverse Flow (IF), a generative modeling framework for inverse generation problems such as denoising without ground truth, and two methods Inverse Flow Match (IFM) and Inverse Consistency Model (ICM) to solve the inverse flow problem. Our framework connects the family of continuous-time generative models to inverse generation problems. Practically, we extended the applicability of denoising without ground truth to almost any continuous noise distributions. We demonstrated strong empirical results applying inverse flow. A limitation of inverse flow is assuming prior knowledge of the noise distribution, and future work is needed to relax this assumption. We expect inverse flow to open up possibilities to explore additional connections to the expanding family of continuous-time generative model methods, and the generalized consistency training objective will expand the application of consistency models.

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

# A   APPENDIX

## A.1   ALTERNATIVE FORMS OF IFM AND ICM

Here we provide the details of alternative objectives and corresponding algorithms of IFM and ICM which are easier and flexible to use.

### A.1.1   ALTERNATIVE OBJECTIVES OF IFM AND ICM

We define the alternative objective of IFM similar to conditional flow matching (Eq. 3):

$$\mathcal{L}_{\text{IFM}}(\theta) = \mathbb{E}_{t,p(\mathbf{x}_1),p\left(\mathbf{x}_1'|\mathbf{x}_0=\text{ODE}_{1\to0}^{\mathbf{v}^\theta}(\mathbf{x}_1)\right),p(\mathbf{x}_t|\mathbf{x}_0,\mathbf{x}_1')} \left[\left\|\mathbf{v}_t^\theta(\mathbf{x}_t) - \mathbf{u}_t\left(\mathbf{x}_t \mid \text{ODE}_{1\to0}^{\mathbf{v}^\theta}(\mathbf{x}_1), \mathbf{x}_1'\right)\right\|\right]$$
(14)

where $\mathbf{x}_1'$ is sampled from the conditional noise distribution. As described in Section 2.1.1 $\mathbf{u}_t(\mathbf{x} \mid \mathbf{x}_0, \mathbf{x}_1')$ can be easily chosen as any smooth interpolation between $\mathbf{x}_0$ and $\mathbf{x}_1'$, such as $\mathbf{u}_t(\mathbf{x} \mid \mathbf{x}_0, \mathbf{x}_1') = \mathbf{x}_1' - \mathbf{x}_0$.

Since ICM is based on generalized consistency training, we first provide the alternative objective of generalized consistency training

$$\mathcal{L}_{\text{GCT}}(\theta) = \mathbb{E}_{i,p(\mathbf{x}_0,\mathbf{x}_1),p(\mathbf{x}_{t_{i+1}}|\mathbf{x}_0,\mathbf{x}_1)} \left[\left\|\mathbf{c}_\theta(\mathbf{x}_{t_{i+1}}, t_{i+1}) - \text{stopgrad}\left(\mathbf{c}_\theta(\mathbf{x}_{t_i}, t_i)\right)\right\|\right],$$
$$\mathbf{x}_{t_{i+1}} - \mathbf{x}_{t_i} = \mathbf{u}_{t_{i+1}}(\mathbf{x}_{t_{i+1}} \mid \mathbf{x}_0, \mathbf{x}_1)(t_{i+1} - t_i)$$
(15)

where the conditional flow is defined jointly by $p(\mathbf{x}_1 \mid \mathbf{x}_0)$ and $\mathbf{u}_{t_{i+1}}(\mathbf{x} \mid \mathbf{x}_0, \mathbf{x}_1)$.

Then the alterntive form of ICM can be defined as

$$\mathcal{L}_{\text{ICM}}(\theta) =$$
$$\mathbb{E}_{i,p(\mathbf{x}_1),p(\mathbf{x}_1'|\mathbf{x}_0=\mathbf{c}_\theta(\mathbf{x}_1,1)),p(\mathbf{x}_{t_{i+1}}|\mathbf{x}_0=\mathbf{c}_\theta(\mathbf{x}_1,1),\mathbf{x}_1')} \left[\left\|\mathbf{c}_\theta(\mathbf{x}_{t_{i+1}}, t_{i+1}) - \text{stopgrad}\left(\mathbf{c}_\theta(\mathbf{x}_{t_i}, t_i)\right)\right\|\right],$$
$$\mathbf{x}_{t_{i+1}} - \mathbf{x}_{t_i} = \mathbf{u}_{t_{i+1}}(\mathbf{x}_{t_{i+1}} \mid \mathbf{x}_0, \mathbf{x}_1')(t_{i+1} - t_i)$$
(16)

where $\mathbf{u}_t(\mathbf{x} \mid \mathbf{x}_0, \mathbf{x}_1')$ can be freely defined based on any interpolation between $\mathbf{x}_0$ and $\mathbf{x}_1'$, which is more easily applicable to any conditional noise distribution:.

### A.1.2   ALTERNATIVE ALGORITHMS OF IFM AND ICM

Here we show the algorithms of alternative objectives of IFM (Eq. 14) and ICM (Eq. 16).

---

**Algorithm 3** IFM Training v2.

1: **Input:** dataset $\mathcal{D}$, initial model parameter $\theta$, and learning rate $\eta$
2: **repeat**
3:     Sample $\mathbf{x}_1 \sim \mathcal{D}$ and $t \sim \mathcal{U}[0,1]$
4:     $\mathbf{x}_0 \leftarrow \text{stopgrad}\left(\text{ODE}_{1\to0}^{\mathbf{v}^\theta}(\mathbf{x}_1)\right)$
5:     Sample $\mathbf{x}_1' \sim p(\mathbf{x}_1' \mid \mathbf{x}_0)$
6:     Sample $\mathbf{x}_t \sim p(\mathbf{x}_t \mid \mathbf{x}_0, \mathbf{x}_1')$
7:     $\mathcal{L}(\theta) \leftarrow$
        $\left\|\mathbf{v}_t^\theta(\mathbf{x}_t) - \mathbf{u}_t(\mathbf{x}_t \mid \mathbf{x}_0, \mathbf{x}_1')\right\|^2$
8:     $\theta \leftarrow \theta - \eta\nabla_\theta\mathcal{L}(\theta)$
9: **until** convergence

---

**Algorithm 4** ICM Training v2.

1: **Input:** dataset $\mathcal{D}$, initial model parameter $\theta$, learning rate $\eta$, and sequence of time points $0 = t_1 < t_2 < \cdots < t_N = 1$
2: **repeat**
3:     Sample $\mathbf{x}_1 \sim \mathcal{D}$ and $i \sim \mathcal{U}[1, N-1]$
4:     $\mathbf{x}_0 \leftarrow \text{stopgrad}(\mathbf{c}_\theta(\mathbf{x}_1, 1))$
5:     Sample $\mathbf{x}_1' \sim p(\mathbf{x}_1' \mid \mathbf{x}_0)$
6:     Sample $\mathbf{x}_{t_{i+1}} \sim p(\mathbf{x}_{t_{i+1}} \mid \mathbf{x}_0, \mathbf{x}_1')$
7:     $\mathbf{x}_{t_i} \leftarrow$
        $\mathbf{x}_{t_{i+1}} - \mathbf{u}_{t_{i+1}}(\mathbf{x}_{t_{i+1}} \mid \mathbf{x}_0, \mathbf{x}_1')(t_{i+1} - t_i)$
8:     $\mathcal{L}(\theta) \leftarrow$
        $d\left[\mathbf{c}_\theta(\mathbf{x}_{t_{i+1}}, t_{i+1}), \text{stopgrad}\left(\mathbf{c}_\theta(\mathbf{x}_{t_i}, t_i)\right)\right]$
9:     $\theta \leftarrow \theta - \eta\nabla_\theta\mathcal{L}(\theta)$
10: **until** convergence

---

## A.2 PROOFS

### A.2.1 INVERSE FLOW MATCHING

**Theorem 1:** Assume that the conditional noise distribution $p(\mathbf{x}_1 \mid \mathbf{x}_0)$ satisfies the condition that, for any noisy data distribution $p(\mathbf{x}_1)$ there exists only one probability distribution $p(\mathbf{x}_0)$ that satisfies $p(\mathbf{x}_1) = \int p(\mathbf{x}_1 \mid \mathbf{x}_0)p(\mathbf{x}_0)\mathrm{d}\mathbf{x}_0$ , then under the condition that $\mathcal{L}_{\mathrm{IFM}}$ is minimized, we have $q(\mathbf{x}_0) = p(\mathbf{x}_0)$.

*Proof:*

The inferred data distribution is given by the push-forward operator (Lipman et al., 2023):

$$q(\mathbf{x}_0) = \left[\mathrm{ODE}_{1\to 0}^{\mathbf{v}^\theta}\right] * p(\mathbf{x}_1) \tag{17}$$

which is defined for any continuous normalizing flow $\phi$ from $\mathbf{x}_1$ to $\mathbf{x}_0$ in the form of

$$[\phi] * p(\mathbf{x}_1) = p\left(\phi^{-1}(\mathbf{x}_0)\right) \det\left[\frac{\partial \phi^{-1}}{\partial \mathbf{x}}(\mathbf{x}_0)\right] \tag{18}$$

where $\mathbf{x}_1 = \phi^{-1}(\mathbf{x}_0)$. The inferred noisy data distribution $q(\mathbf{x}_1)$ is given by

$$q(\mathbf{x}_1) = \int p(\mathbf{x}_1 \mid \mathbf{x}_0)q(\mathbf{x}_0)\mathrm{d}\mathbf{x}_0 \tag{19}$$

When the model is converged based on the condition $\mathcal{L}_{\mathrm{IFM}}$ is minimized, we have

$$q(\mathbf{x}_0) = \left[\mathrm{ODE}_{1\to 0}^{\mathbf{v}^\theta}\right] * q(\mathbf{x}_1) \tag{20}$$

Then we find that

$$\left[\mathrm{ODE}_{1\to 0}^{\mathbf{v}^\theta}\right] * p(\mathbf{x}_1) = \left[\mathrm{ODE}_{1\to 0}^{\mathbf{v}^\theta}\right] * q(\mathbf{x}_1) \tag{21}$$

By the definition of the push-forward operator, we have

$$p\left(\left(\mathrm{ODE}_{1\to 0}^{\mathbf{v}^\theta}\right)^{-1}(\mathbf{x}_0)\right) \det\left[\frac{\partial \left(\mathrm{ODE}_{1\to 0}^{\mathbf{v}^\theta}\right)^{-1}}{\partial \mathbf{x}}(\mathbf{x}_0)\right]$$
$$= q\left(\left(\mathrm{ODE}_{1\to 0}^{\mathbf{v}^\theta}\right)^{-1}(\mathbf{x}_0)\right) \det\left[\frac{\partial \left(\mathrm{ODE}_{1\to 0}^{\mathbf{v}^\theta}\right)^{-1}}{\partial \mathbf{x}}(\mathbf{x}_0)\right] \tag{22}$$

Since the solution of ODE is unique, $\mathrm{ODE}_{1\to 0}^{\mathbf{v}^\theta}$ is a bijective function with

$$\left(\mathrm{ODE}_{1\to 0}^{\mathbf{v}^\theta}\right)^{-1} = \mathrm{ODE}_{0\to 1}^{\mathbf{v}^\theta}$$

and

$$\mathbf{x}_1 = \mathrm{ODE}_{0\to 1}^{\mathbf{v}^\theta}(\mathbf{x}_0) = \left(\mathrm{ODE}_{1\to 0}^{\mathbf{v}^\theta}\right)^{-1}(\mathbf{x}_0)$$

Also, the nontrivial solution ensures that the determinant is non-zero. By substitution, we get

$$p(\mathbf{x}_1) = q(\mathbf{x}_1) \tag{23}$$

and combine with Eq. 19, we find that

$$p(\mathbf{x}_1) = \int p(\mathbf{x}_1 \mid \mathbf{x}_0)q(\mathbf{x}_0)\mathrm{d}\mathbf{x}_0 \tag{24}$$

We close the proof by directly applying the uniqueness of $p(\mathbf{x}_0)$ and find that

$$q(\mathbf{x}_0) = p(\mathbf{x}_0) \tag{25}$$

**Remark 1:** Readers may notice that if $q(x_0)$ is a point mass, which means the model maps all inputs to a constant, the training objective $\mathcal{L}_{\mathrm{IFM}}$ will also be minimized, causing the ODE to converge to a trivial solution. However, we find that this trivial solution can be avoided by our design. Specifically, this is because Our approach can be regarded as an optimization process based on expectation-maximization (EM):

1. **Expectation:** We generate a denoised dataset given noisy inputs, $x_0 \sim q(x_0|x_1)$.
2. **Maximization:** We optimize our IFM/ICM models based on the generated dataset $x_0$ and the conditional noise distribution $p(x_1|x_0)$.

The choice of the initial prior, which is the initial denoised dataset in our case, is crucial for the EM algorithm. While any initial prior may lead to a local optimum (Wu, 1983; Balakrishnan et al., 2014; McLachlan & Krishnan, 2008), an informed initial prior can prevent convergence to a trivial solution. Our model architecture incorporates the residual connection from consistency models, ensuring that the initial outputs of the model closely resemble the inputs. This design effectively avoids convergence to the trivial solution. In additional experiments (Appendix A.5.1), we further demonstrate that our method is able to converge even under high noise levels ($\sigma = 50$), corroborating the reliability of our method.

Therefore, when the training objective converges, our proof remains valid since the one-to-one mapping property of the ODE holds.

Our method shares similarities with EM-based diffusion (Rozet et al., 2024). However, our method exhibits greater versatility by being applicable to removing various types of noise. Moreover, the design of ICM, inspired by consistency models, eliminates the need for multi-step ODE sampling during training and inference, resulting in a significantly faster process.

**Lemma 1:** Given a conditional ODE vector field $\mathbf{u}_t(\mathbf{x} \mid \mathbf{x}_0, \mathbf{x}_1)$ that generates a conditional probability path $p(\mathbf{x}_t \mid \mathbf{x}_0, \mathbf{x}_1)$, the unconditional probability path $p(\mathbf{x}_t)$ can be generated by the unconditional ODE vector field $\mathbf{u}_t(\mathbf{x})$, which is defined as

$$\mathbf{u}_t(\mathbf{x}) = \mathbb{E}_{p(\mathbf{x}_0, \mathbf{x}_1|\mathbf{x})} \left[ \mathbf{u}_t(\mathbf{x} \mid \mathbf{x}_0, \mathbf{x}_1) \right] \tag{26}$$

*Proof:*

To verify this, we check that $p(\mathbf{x}_t)$ and $\mathbf{u}_t(\mathbf{x})$ satisfy the continuity equation:

$$\frac{d}{dt} p(\mathbf{x}_t) + \mathrm{div}\left( \mathbf{u}_t(\mathbf{x}) p(\mathbf{x}_t) \right) = 0. \tag{27}$$

By definition,

$$\frac{d}{dt} p(\mathbf{x}_t) = \frac{d}{dt} \int p(\mathbf{x}_t|\mathbf{x}_0, \mathbf{x}_1) p(\mathbf{x}_0, \mathbf{x}_1) \mathrm{d}\mathbf{x}_0 \mathrm{d}\mathbf{x}_1. \tag{28}$$

With Leibniz Rule we have

$$\frac{d}{dt} p(\mathbf{x}_t) = \int \frac{d}{dt} p(\mathbf{x}_t|\mathbf{x}_0, \mathbf{x}_1) p(\mathbf{x}_0, \mathbf{x}_1) \mathrm{d}\mathbf{x}_0 \mathrm{d}\mathbf{x}_1. \tag{29}$$

Since $\mathbf{u}_t(\mathbf{x}|\mathbf{x}_0, \mathbf{x}_1)$ generates $p(\mathbf{x}_t|\mathbf{x}_0, \mathbf{x}_1)$, by the continuity equation we have

$$\frac{d}{dt} p(\mathbf{x}_t|\mathbf{x}_0, \mathbf{x}_1) + \mathrm{div}\left( \mathbf{u}_t(\mathbf{x}|\mathbf{x}_0, \mathbf{x}_1) p(\mathbf{x}_t|\mathbf{x}_0, \mathbf{x}_1) \right) = 0. \tag{30}$$

Substitution in Eq. 29 gives

$$\frac{d}{dt} p(\mathbf{x}_t) = - \int \mathrm{div}\left( \mathbf{u}_t(\mathbf{x}|\mathbf{x}_0, \mathbf{x}_1) p(\mathbf{x}_t|\mathbf{x}_0, \mathbf{x}_1) \right) p(\mathbf{x}_0, \mathbf{x}_1) \mathrm{d}\mathbf{x}_0 \mathrm{d}\mathbf{x}_1. \tag{31}$$

Exchanging the derivative and integral,

$$\frac{d}{dt} p(\mathbf{x}_t) = -\mathrm{div} \int \left( \mathbf{u}_t(\mathbf{x}|\mathbf{x}_0, \mathbf{x}_1) p(\mathbf{x}_t|\mathbf{x}_0, \mathbf{x}_1) p(\mathbf{x}_0, \mathbf{x}_1) \mathrm{d}\mathbf{x}_0 \mathrm{d}\mathbf{x}_1 \right). \tag{32}$$

The definition of $\mathbf{u}_t(\mathbf{x})$ is

$$\mathbf{u}_t(\mathbf{x}) = \mathbb{E}_{p(\mathbf{x}_0, \mathbf{x}_1|\mathbf{x})} \left[ \mathbf{u}_t(\mathbf{x} \mid \mathbf{x}_0, \mathbf{x}_1) \right] = \int \mathbf{u}_t(\mathbf{x} \mid \mathbf{x}_0, \mathbf{x}_1) \frac{p(\mathbf{x}_t|\mathbf{x}_0, \mathbf{x}_1) p(\mathbf{x}_0, \mathbf{x}_1)}{p(\mathbf{x}_t)}. \tag{33}$$

Combining Eq. 32 and Eq. 33 gives the continuity equation:

$$\frac{d}{dt} p(\mathbf{x}_t) + \mathrm{div}\left( \mathbf{u}_t(\mathbf{x}) p(\mathbf{x}_t) \right) = 0. \tag{34}$$

### A.2.2 GENERALIZED CONSISTENCY TRAINING

Without loss of generality, we provide the proof for the form of $\mathcal{L}_{\mathrm{GCT}}$ in Eq. 15, and the proof for the form Eq. 10 follows by assuming that the forward conditional probability path is independent of $\mathbf{x}_1$.

**Theorem 2**: Assuming the consistency function $\mathbf{c}_\theta$ is twice differentiable, up to a constant independent of $\theta$, $\mathcal{L}_{\mathrm{GCT}}$ and $\mathcal{L}_{\mathrm{CD}}$ are equal.

*Proof:*

The proof is inspired by Song et al. (2023). We use the shorthand $\mathbf{c}_{\theta-}$ to denote the stopgrad version of the consistency function $\mathbf{c}$. Given a multi-variate function $\mathbf{h}(\mathbf{x}, \mathbf{y})$, the operator $\partial_1 \mathbf{h}(\mathbf{x}, \mathbf{y})$ and $\partial_2 \mathbf{h}(\mathbf{x}, \mathbf{y})$ denote the partial derivative with respect to $\mathbf{x}$ and $\mathbf{y}$. Let $\Delta t := \max_i \{| t_{i+1} - t_i |\}$ and we use $o(\Delta t)$ to denote infinitesimal with respect to $\Delta t$.

Based on Eq. 5 and Eq. 4, the consistency distillation objective is

$$\mathcal{L}_{\mathrm{CD}}(\theta) = \mathbb{E}_{i, p(\mathbf{x}_0, \mathbf{x}_1), p(\mathbf{x}_{t_{i+1}} | \mathbf{x}_0, \mathbf{x}_1)} \left\{ d \left[ \mathbf{c}_\theta(\mathbf{x}_{t_{i+1}}, t_{i+1}), \mathbf{c}_{\theta-}(\mathbf{x}_{t_i}, t_i) \right] \right\} \tag{35}$$

where $\mathbf{x}_{t_i} = \mathbf{x}_{t_{i+1}} - (t_{i+1} - t_i) \mathbf{u}_{t_{i+1}}(\mathbf{x}_{t_{i+1}})$ and $d$ is a general distance function.

We assume $d$ and $\mathbf{c}_{\theta-}$ are twice continuously differentiable with bounded derivatives. With Taylor expansion, we have

$$
\begin{aligned}
\mathcal{L}_{\mathrm{CD}}(\theta) &= \mathbb{E}_{i, p(\mathbf{x}_0, \mathbf{x}_1), p(\mathbf{x}_{t_{i+1}} | \mathbf{x}_0, \mathbf{x}_1)} \left\{ d \left[ \mathbf{c}_\theta(\mathbf{x}_{t_{i+1}}, t_{i+1}), \mathbf{c}_{\theta-}(\mathbf{x}_{t_i}, t_i) \right] \right\} \\
&= \mathbb{E}_{i, p(\mathbf{x}_0, \mathbf{x}_1), p(\mathbf{x}_{t_{i+1}} | \mathbf{x}_0, \mathbf{x}_1)} \left\{ d \left[ \mathbf{c}_\theta(\mathbf{x}_{t_{i+1}}, t_{i+1}), \mathbf{c}_{\theta-}(\mathbf{x}_{t_{i+1}} - (t_{i+1} - t_i) \mathbf{u}_{t_{i+1}}(\mathbf{x}_{t_{i+1}}), t_i) \right] \right\} \\
&= \mathbb{E}_{i, p(\mathbf{x}_0, \mathbf{x}_1), p(\mathbf{x}_{t_{i+1}} | \mathbf{x}_0, \mathbf{x}_1)} \left\{ d \left[ \mathbf{c}_\theta(\mathbf{x}_{t_{i+1}}, t_{i+1}), \mathbf{c}_{\theta-}(\mathbf{x}_{t_{i+1}}, t_{i+1}) \right. \right. \\
&\quad - \partial_1 \mathbf{c}_{\theta-}(\mathbf{x}_{t_{i+1}}, t_{i+1})(t_{i+1} - t_i) \mathbf{u}_{t_{i+1}}(\mathbf{x}_{t_{i+1}}) \\
&\quad \left. \left. - \partial_2 \mathbf{c}_{\theta-}(\mathbf{x}_{t_{i+1}}, t_{i+1})(t_{i+1} - t_i) + o(\Delta t) \right] \right\} \\
&= \mathbb{E}_{i, p(\mathbf{x}_0, \mathbf{x}_1), p(\mathbf{x}_{t_{i+1}} | \mathbf{x}_0, \mathbf{x}_1)} \left\{ d \left[ \mathbf{c}_\theta(\mathbf{x}_{t_{i+1}}, t_{i+1}), \mathbf{c}_{\theta-}(\mathbf{x}_{t_{i+1}}, t_{i+1}) \right] \right\} \\
&\quad - \mathbb{E}_{i, p(\mathbf{x}_0, \mathbf{x}_1), p(\mathbf{x}_{t_{i+1}} | \mathbf{x}_0, \mathbf{x}_1)} \left\{ \partial_2 d \left[ \mathbf{c}_\theta(\mathbf{x}_{t_{i+1}}, t_{i+1}), \mathbf{c}_{\theta-}(\mathbf{x}_{t_{i+1}}, t_{i+1}) \right] \right. \\
&\quad \left. \cdot \left[ \partial_1 \mathbf{c}_{\theta-}(\mathbf{x}_{t_{i+1}}, t_{i+1})(t_{i+1} - t_i) \mathbf{u}_{t_{i+1}}(\mathbf{x}_{t_{i+1}}) \right] \right\} \\
&\quad - \mathbb{E}_{i, p(\mathbf{x}_0, \mathbf{x}_1), p(\mathbf{x}_{t_{i+1}} | \mathbf{x}_0, \mathbf{x}_1)} \left\{ \partial_2 d \left[ \mathbf{c}_\theta(\mathbf{x}_{t_{i+1}}, t_{i+1}), \mathbf{c}_{\theta-}(\mathbf{x}_{t_{i+1}}, t_{i+1}) \right] \right. \\
&\quad \left. \cdot \left[ \partial_2 \mathbf{c}_{\theta-}(\mathbf{x}_{t_{i+1}}, t_{i+1})(t_{i+1} - t_i) \right] \right\} + \mathbb{E} \left[ o(\Delta t) \right]
\end{aligned}
\tag{36}
$$

Then, we apply Lemma 1 and use Taylor expansion in the reverse direction,

$$
\begin{aligned}
&\mathcal{L}_{\mathrm{CD}}(\theta) \\
&= \mathbb{E}_{i, p(\mathbf{x}_0, \mathbf{x}_1), p(\mathbf{x}_{t_{i+1}} | \mathbf{x}_0, \mathbf{x}_1)} \left\{ d \left[ \mathbf{c}_\theta(\mathbf{x}_{t_{i+1}}, t_{i+1}), \mathbf{c}_{\theta-}(\mathbf{x}_{t_{i+1}}, t_{i+1}) \right] \right\} \\
&\quad - \mathbb{E}_{i, p(\mathbf{x}_0, \mathbf{x}_1), p(\mathbf{x}_{t_{i+1}} | \mathbf{x}_0, \mathbf{x}_1)} \left\{ \partial_2 d \left[ \mathbf{c}_\theta(\mathbf{x}_{t_{i+1}}, t_{i+1}), \mathbf{c}_{\theta-}(\mathbf{x}_{t_{i+1}}, t_{i+1}) \right] \right. \\
&\quad \left. \cdot \left[ \partial_1 \mathbf{c}_{\theta-}(\mathbf{x}_{t_{i+1}}, t_{i+1})(t_{i+1} - t_i) \mathbb{E}_{p(\mathbf{x}_0, \mathbf{x}_1 | \mathbf{x}_{t_{i+1}})} \left[ \mathbf{u}_{t_{i+1}}(\mathbf{x}_{t_{i+1}} | \mathbf{x}_0, \mathbf{x}_1) \right] \right] \right\} \\
&\quad - \mathbb{E}_{i, p(\mathbf{x}_0, \mathbf{x}_1), p(\mathbf{x}_{t_{i+1}} | \mathbf{x}_0, \mathbf{x}_1)} \left\{ \partial_2 d \left[ \mathbf{c}_\theta(\mathbf{x}_{t_{i+1}}, t_{i+1}), \mathbf{c}_{\theta-}(\mathbf{x}_{t_{i+1}}, t_{i+1}) \right] \right. \\
&\quad \left. \cdot \left[ \partial_2 \mathbf{c}_{\theta-}(\mathbf{x}_{t_{i+1}}, t_{i+1})(t_{i+1} - t_i) \right] \right\} + \mathbb{E} \left[ o(\Delta t) \right] \\
&\stackrel{(i)}{=} \mathbb{E}_{i, p(\mathbf{x}_0, \mathbf{x}_1), p(\mathbf{x}_{t_{i+1}} | \mathbf{x}_0, \mathbf{x}_1)} \left\{ d \left[ \mathbf{c}_\theta(\mathbf{x}_{t_{i+1}}, t_{i+1}), \mathbf{c}_{\theta-}(\mathbf{x}_{t_{i+1}}, t_{i+1}) \right] \right\} \\
&\quad - \mathbb{E}_{i, p(\mathbf{x}_0, \mathbf{x}_1), p(\mathbf{x}_{t_{i+1}} | \mathbf{x}_0, \mathbf{x}_1)} \left\{ \partial_2 d \left[ \mathbf{c}_\theta(\mathbf{x}_{t_{i+1}}, t_{i+1}), \mathbf{c}_{\theta-}(\mathbf{x}_{t_{i+1}}, t_{i+1}) \right] \right. \\
&\quad \left. \cdot \left[ \partial_1 \mathbf{c}_{\theta-}(\mathbf{x}_{t_{i+1}}, t_{i+1})(t_{i+1} - t_i) \mathbf{u}_{t_{i+1}}(\mathbf{x}_{t_{i+1}} | \mathbf{x}_0, \mathbf{x}_1) \right] \right\} \\
&\quad - \mathbb{E}_{i, p(\mathbf{x}_0, \mathbf{x}_1), p(\mathbf{x}_{t_{i+1}} | \mathbf{x}_0, \mathbf{x}_1)} \left\{ \partial_2 d \left[ \mathbf{c}_\theta(\mathbf{x}_{t_{i+1}}, t_{i+1}), \mathbf{c}_{\theta-}(\mathbf{x}_{t_{i+1}}, t_{i+1}) \right] \right. \\
&\quad \left. \cdot \left[ \partial_2 \mathbf{c}_{\theta-}(\mathbf{x}_{t_{i+1}}, t_{i+1})(t_{i+1} - t_i) \right] \right\} + \mathbb{E} \left[ o(\Delta t) \right] \\
&= \mathbb{E}_{i, p(\mathbf{x}_0, \mathbf{x}_1), p(\mathbf{x}_{t_{i+1}} | \mathbf{x}_0, \mathbf{x}_1)} \left\{ d \left[ \mathbf{c}_\theta(\mathbf{x}_{t_{i+1}}, t_{i+1}), \mathbf{c}_{\theta-}(\mathbf{x}_{t_{i+1}}, t_{i+1}) \right. \right.
\end{aligned}
$$

$$
\begin{aligned}
&\quad - \partial_1 \mathbf{c}_{\theta^-}(\mathbf{x}_{t_{i+1}}, t_{i+1})(t_{i+1} - t_i)\mathbf{u}_{t_{i+1}}(\mathbf{x}_{t_{i+1}} \mid \mathbf{x}_0, \mathbf{x}_1) \\
&\qquad - \partial_2 \mathbf{c}_{\theta^-}(\mathbf{x}_{t_{i+1}}, t_{i+1})(t_{i+1} - t_i) + o(\Delta t) \big] \big\} \\
&= \mathbb{E}_{i, p(\mathbf{x}_0, \mathbf{x}_1), p(\mathbf{x}_{t_{i+1}} | \mathbf{x}_0, \mathbf{x}_1)} \left\{ d \left[ \mathbf{c}_\theta\left(\mathbf{x}_{t_{i+1}}, t_{i+1}\right), \mathbf{c}_{\theta^-}(\mathbf{x}_{t_{i+1}} - (t_{i+1} - t_i)\mathbf{u}_{t_{i+1}}(\mathbf{x}_{t_{i+1}} \mid \mathbf{x}_0, \mathbf{x}_1), t_i) \right] \right\} \\
&\qquad + o(\Delta t) \\
&= \mathcal{L}_{\mathrm{GCT}}(\theta) + o(\Delta t)
\end{aligned}
\tag{37}
$$

where (i) is due to the law of total expectation.

## A.3 Introduction to denoising without ground truth

The most comparable approaches to our method are those that explicitly consider a noise distribution, including Stein's Unbiased Risk Estimate (SURE)-based denoising methods (Soltanayev & Chun, 2018; Metzler et al., 2020) and Noise2Score (Kim & Ye, 2021). SURE-based denoising is applicable to independent Gaussian noise and Noise2Score is more generally applicable to exponential family noise. SURE-based denoising directly optimizes a loss motivated by SURE which provides an unbiased estimate of the true risk, which is a mean-squared error to the ground truth. Noise2Score uses Tweedie's formula for estimating the posterior mean of an exponential family distribution with the score of the noisy distribution. The score is estimated by an approximate score estimator using a denoising autoencoder.

Another family of approaches often referred to as Noise2X is based on the assumptions of centered (zero-mean) and independent noise. Noise2Noise (Lehtinen et al., 2018) requires independent noisy observations of the same ground truth data. Noise2Self (Batson & Royer, 2019) is based on the statistical independence across different dimensions of the measurement, such as the independence between different pixels. Noise2Void (Krull et al., 2019) leverages the concept of blind-spot networks, which predict the value of a pixel based solely on its surrounding context. Similarly, Noise2Same (Xie et al., 2020) employs self-supervised learning using selectively masked or perturbed regions to train the model to predict unobserved values. Both of them assume independence of noise across dimensions.

## A.4 Experimental details

All experiments were conducted on a server with 36 cores, 400 GB memory, and NVIDIA Tesla V100 GPUs. All models were implemented with PyTorch 2.1 (Paszke et al., 2019) and trained with the AdamW (Loshchilov & Hutter, 2019) optimizer. Model architectures and training hyperparameters are listed in Table A.4.

Table 2: Model architectures and hyperparameters

| dataset | architecture | channels | embed_dim | embed_scale | epochs | lr | lr schedule |
|---|---|---|---|---|---|---|---|
| Navier-Stokes | | | | | 2000 | $5 \times 10^{-4}$ | |
| 8-gaussians | MLP | [256,256, 256,256] | 256 | 1.0 | 2000 | $5 \times 10^{-4}$ | None |
| Single-cell | | | | | 1000 | $1 \times 10^{-4}$ | |
| Gaussian noise | | | | | 3000 | $1 \times 10^{-4}$ | StepLR |
| Correlated noise | UNet | [128,128, 256,256,512] | 512 | 1.0 | 1000 | $1 \times 10^{-4}$ | None |
| Jacobi process | | | | | 1000 | $1 \times 10^{-4}$ | None |
| FMD | | | | | 3000 | $1 \times 10^{-4}$ | StepLR |

### A.4.1 Training details

To train IFM or ICM, we first consider a discretized time sequence $\epsilon = t_1 < t_2 < \cdots < t_N = 1$, where $\epsilon$ is a small positive value close to 0. We follow Karras et al. (2022) to determine the time sequence with the formula $t_i = \left( \epsilon^{1/\rho} + \frac{i-1}{N-1}(T^{1/\rho} - \epsilon^{1/\rho}) \right)^\rho$, where $\rho = 7$, $T = 1$, and $N = 11$. We choose the conditional ODE vector field as

$$
\mathbf{u}_{t_i}(\mathbf{x}_{t_i} \mid \mathbf{x}_0, \mathbf{x}_1) = \mathbf{x}_1 - \mathbf{x}_0.
\tag{38}
$$

Further, the gradient of the inferred noise-free data $\mathbf{x}_0$ is stopped to stabilize the training process, which is

$$\mathbf{x}_0 = \text{stopgrad}\left(\text{ODE}_{1\to0}^{\mathbf{v}^\theta}(\mathbf{x}_1)\right) \tag{39}$$

for IFM and

$$\mathbf{x}_0 = \text{stopgrad}\left(\mathbf{c}_\theta(\mathbf{x}_1, 1)\right) \tag{40}$$

for ICM. For ICM, the loss is weighted by

$$\lambda(i) = t_{i+1} - t_i \tag{41}$$

in the same way as Song & Dhariwal (2023).

### A.4.2  SYNTHETIC DATASETS

We adopted a simple form of Navier-Stokes equations which only includes the viscosity term in the fluid mechanics

$$\rho(\frac{\partial v}{\partial t} + v \cdot \nabla v) = -\nabla p + \mu \nabla^2 v$$
$$\nabla \cdot v = 0 \tag{42}$$

where $\rho$ is the density of the fluid, $v$ is the velocity, $p$ is the pressure and $\mu$ is the viscosity coefficient. For inverting the Navier-Stokes simulations, we simulated the fluid data within a 2D boundary of $[0,1] \times [0,1]$ domain from $t = 0$ to $t = 0.1$ with the spectral method (Spalart et al., 1991)

The 8-gaussians is generated by adding independent gaussian noise ($\sigma = 0.15$) to 8 points whose coordinates are $(0,1), (0. - 1), (1,0), (-1,0), (\frac{\sqrt{2}}{2}, \frac{\sqrt{2}}{2}), (\frac{\sqrt{2}}{2}, -\frac{\sqrt{2}}{2}), (-\frac{\sqrt{2}}{2}, \frac{\sqrt{2}}{2}), (-\frac{\sqrt{2}}{2}, -\frac{\sqrt{2}}{2})$. The dataset is composed of 8000 points for training and 1600 points for testing.

We used a simple MLP-based model architecture with Gaussian Fourier time embedding in Table A.4. All methods were trained with a learning rate of $5 \times 10^{-4}$ for 2000 epochs. The model training took about 10 minutes.

### A.4.3  REAL-WORLD DATASETS

All models were trained using the BSDS500 training set with 200 images randomly cropped to the size of $256 \times 256$ and evaluated on the BSDS500 test set, Kodak, and Set12 with images cropped to the same size at the center. We used the same UNet-based model architecture as Lehtinen et al. (2018) with additional Gaussian Fourier time embedding listed in Table A.4.

The URL for each dataset is given:

BSDS500 (Arbeláez et al., 2011): `https://www2.eecs.berkeley.edu/Research/Projects/CS/vision/bsds/`

Kodak: `https://r0k.us/graphics/kodak/`

Set12 (Zhang et al., 2017): `https://github.com/cszn/DnCNN/tree/master/testsets/Set12`

**Gaussian noise** is applied with

$$\mathbf{x}_1 = \mathbf{x}_0 + \eta \tag{43}$$

where $\mathbf{x}_0$ is the noise-free data, $\mathbf{x}_1$ is a noisy observation, and $\eta \sim \mathcal{N}(0, \sigma^2 I)$. We chose $\sigma = 25$ in the experiments. All models were trained with the following setting. The total epoch was set to 3000. The learning rate was initialized to $1 \times 10^{-4}$ for the first 1500 epochs and was decayed to $5 \times 10^{-5}$ for the last 1500 epochs. The model training took about 1.5 hours.

**Correlated noise** is applied similarly to independent Gaussian noise. We adopt the method from Mäkinen et al. (2020) with

$$\eta = \nu \circledast g \tag{44}$$

where $\nu \sim \mathcal{N}(0, \sigma^2 I)$ and $g$ is a convolution kernel. We consider $g$ in the form of

$$g = \frac{1}{2\pi a^2} \cos|r| \exp\left(-\frac{r^2}{2a^2}\right) \tag{45}$$

in polar coordinates and $a$ determines the level of correlation. We generated the correlated noisy observation with $\sigma = 25$ and $a = 2$. All models were trained with a learning rate of $1 \times 10^{-4}$ for 1000 epochs. The model training took about 30 minutes.

**Jacobi process** takes the following form

$$d\mathbf{x} = \frac{s}{2}[a(1 - \mathbf{x}) - b\mathbf{x}]dt + \sqrt{s\mathbf{x}(1 - \mathbf{x})}d\mathbf{w}, \tag{46}$$

where $0 \leq x \leq 1$, $s > 0$ is the speed factor, and $a > 0$, $b > 0$ determines the stationary distribution $\text{Beta}(a, b)$. Note that when $x$ approaches 0 or 1, the diffusion coefficient converges to 0 and the drift coefficient converges to $a$ or $-b$, keeping the diffusion within $[0, 1]$. We used $s = 1$ and $a = b = 1$ and generated the noisy observation $\mathbf{x}_1$ with an Euler-Maruyama sampler to simulate the SDE from the initial value $\mathbf{x}_0$. All models were trained with a learning rate of $1 \times 10^{-4}$ for 1000 epochs. The model training took about 1.5 hours.

### A.4.4 DENOISING MICROSCOPIC DATA

The Fluorescence Microscopy Denoising (FMD) dataset published by Zhang et al. (2019) was downloaded from `https://github.com/yinhaoz/denoising-fluorescence`. We adopted the signal dependent noise model from Liu et al. (2013)

$$g = f + f^\gamma \cdot u + w \tag{47}$$

to estimate the condition noise distribution where $g$ is the noisy pixel value, $f$ is the noise-free pixel value, $\gamma$ is the exponential parameter, and $u$ and $w$ are zero-mean random variables with variance $\sigma_u^2$ and $\sigma_w^2$, respectively. The variance of the noise model is

$$\sigma^2 = f^{2\gamma} \cdot \sigma_u^2 + \sigma_w^2. \tag{48}$$

To estimate the parameters in the noise model, we split an image into $4 \times 4$ patches. We assume the variance within a patch is constant and approximate the noise-free pixel values of the patches by the mean values. The parameters in the noise model are estimated by the Maximum-Likelihood method.

We used the same UNet-based model architecture as Lehtinen et al. (2018) with additional Gaussian Fourier time embedding listed in Table A.4. The learning rate was initialized to $1 \times 10^{-4}$ for the first 1500 epochs and was decayed to $5 \times 10^{-5}$ for the last 1500 epochs.

### A.4.5 DENOISING SINGLE-CELL GENOMICS DATA

The adult mouse brain dataset published by Zeisel et al. (2018) was downloaded from `https://www.ncbi.nlm.nih.gov/sra/SRP135960`. The dentate gyrus neurogenesis dataset published by Hochgerner et al. (2018a) was downloaded from `https://www.ncbi.nlm.nih.gov/geo/query/acc.cgi?acc=GSE104323` and the neuron- and glia-related cells were kept for denoising. We preprocessed the datasets by the standard pipeline (Wolf et al., 2018) and then performed principal component analysis. We further normalized the datasets by scaling the standard deviation of the first principal component to 1. After that, we denoised the datasets using the top 6 principal components with $\sigma = 0.4$. We used a simple MLP-based model architecture with Gaussian Fourier time embedding in Table A.4. The model was trained with a learning rate of $1 \times 10^{-4}$ for 1000 epochs. The model training took about 5 minutes.

### A.5 ADDITIONAL EXPERIMENTS

We provide extensive experiments to measure how different levels of Gaussian noise, different noise level assumptions, and different combinations of noises affect performance. We adopted the same model architecture and training strategy as for FMD in Table A.4. .

### A.5.1 DIFFERENT LEVELS OF GAUSSIAN NOISE

We conducted experiments to evaluate the performance of our method under different intensities of Gaussian noise. We performed experiments from $\sigma = 5$ to $\sigma = 50$ and found that our method is robust over all noise levels we applied (Table A.5.1).

Table 3: Denoising performance for different levels of Gaussian noise measured by PSNR in dB

|  | $\sigma = 5$ | | $\sigma = 12.5$ | | $\sigma = 25$ | | $\sigma = 50$ | | $\sigma = 75$ | |
|---|---|---|---|---|---|---|---|---|---|---|
|  | Input | Pred | Input | Pred | Input | Pred | Input | Pred | Input | Pred |
| BSDS500 | 34.15 | 37.56 | 26.19 | 31.85 | 20.17 | 28.16 | 14.15 | 24.98 | 10.63 | 23.33 |
| Kodak | 34.15 | 37.92 | 26.19 | 32.56 | 20.18 | 29.08 | 14.15 | 25.96 | 10.63 | 24.33 |
| Set12 | 34.15 | 37.87 | 26.20 | 32.78 | 20.16 | 29.19 | 14.13 | 25.78 | 10.63 | 23.86 |

Table 4: Denoising performance for different noise distributions measured by PSNR in dB

| Noise type | | Input | Noise2Void | Noise2Self | Noise2Score | Ours (ICM) |
|---|---|---|---|---|---|---|
| Poisson | BSDS500 | 23.78 | 28.29 | 28.52 | **30.53** | 29.91 |
| $\zeta = 0.01$ | Kodak | 23.60 | 28.76 | 29.36 | **31.10** | 30.58 |
|  | Set12 | 23.08 | 30.01 | 29.23 | **30.94** | 30.68 |
| Gamma | BSDS500 | 26.75 | 29.17 | 27.43 | 31.14 | **32.48** |
| $k = 100$ | Kodak | 26.67 | 30.26 | 28.26 | 31.67 | **32.97** |
|  | Set12 | 25.53 | 30.44 | 28.54 | 31.21 | **33.08** |
| Rayleigh | BSDS500 | 14.03 | 28.57 | 14.86 | 30.37 | **30.55** |
| $\sigma = 0.3$ | Kodak | 13.95 | 29.73 | 14.83 | 30.96 | **31.16** |
|  | Set12 | 12.81 | 29.98 | 13.74 | 30.89 | **31.17** |
| | BSDS500 | 22.40 | 26.45 | 27.76 | 28.54 | **29.26** |
| Poisson+Gaussian | Kodak | 22.25 | 27.67 | 28.86 | 29.02 | **30.02** |
|  | Set12 | 21.88 | 27.81 | 29.23 | 29.10 | **30.03** |
| | BSDS500 | 24.29 | 27.98 | 26.10 | 29.34 | **30.53** |
| Gamma+Gaussian | Kodak | 24.24 | 28.99 | 27.08 | 29.90 | **31.22** |
|  | Set12 | 23.62 | 29.53 | 26.84 | 29.69 | **31.27** |
| | BSDS500 | 13.85 | 28.01 | 14.72 | 29.36 | **29.79** |
| Rayleigh+Gaussian | Kodak | 13.77 | 29.12 | 14.69 | 30.12 | **30.49** |
|  | Set12 | 12.78 | 26.81 | 13.59 | 29.82 | **30.50** |
| GaussianRGB | BSDS500 | 20.17 | 29.72 | 27.33 | 28.28 | **29.99** |
| $\sigma = 25$ | Kodak | 20.17 | 30.65 | 28.21 | 28.66 | **30.73** |

### A.5.2 DIFFERENT COMBINATIONS OF NOISES

We considered additive Gaussian noise and multiplicative noise such as Gamma noise, Poisson noise, and Rayleigh noise, as well as their combinations and on a channel-correlated RGB dataset. We followed the noise distributions introduced in Noise2Score (Kim & Ye, 2021; Xie et al., 2023a). For combinations of multiplicative noise and Gaussian noise, we added Gaussian noises with $\sigma = 10$ to the individual multiplicative noise models. As shown in Table A.5.2, our method is robust over all noise type combinations we applied and superior to compared methods in most noise types.

### A.5.3 DIFFERENT NOISE LEVEL ASSUMPTIONS

We conducted experiments on data with $\sigma = 25$ Gaussian noise, but training and denoising with different noise level assumptions from $\sigma = 12.5$ to $\sigma = 50$. Shown in Table A.5.3, our method demonstrates stable performance within the range of $\sigma = 25$ to $\sigma = 35$, indicating that overestimating the noise level has minimal impact on the model's effectiveness.

Table 5: Performance for different noise level assumptions

|  | $\sigma = 12.5$ | $\sigma = 15$ | $\sigma = 20$ | $\sigma = 25$ | $\sigma = 30$ | $\sigma = 35$ | $\sigma = 50$ |
|---|---|---|---|---|---|---|---|
| BSDS500 | 21.59 | 22.43 | 24.78 | **28.16** | 28.09 | 27.55 | 25.71 |
| Kodak | 21.62 | 22.49 | 25.03 | **29.08** | 28.99 | 28.43 | 26.66 |
| Set12 | 21.67 | 22.56 | 25.14 | **29.19** | 29.20 | 28.65 | 26.86 |

### A.5.4 DENOISING SMALL DATASETS

In scientific discovery, the amount of data available is often very limited. To evaluate the performance of our method on small datasets, we conducted experiments on the electron microscopy denoising dataset (Mohan et al., 2021). Since the original authors did not release the real experimental data, we used the simulated dataset they provided and added Poisson noise, which is the noise distribution in the real data according to their analysis. The dataset consists of 46 samples. The results indicate that our method is applicable to small datasets and outperforms other approaches in this scenario (Table A.5.4). While diffusion model is known as being data hungry, our method is efficient on sample size because it does not involve training a full generative model.

Table 6: Performance on the electron microscopy denoising dataset

|      | Input | Noise2Void | Noise2Self | Ours (ICM) |
|------|-------|-----------|-----------|-----------|
| PSNR | 23.70 | 38.67 | 41.42 | **43.78** |

### A.6 ADDITIONAL QUALITATIVE RESULTS

We provide additional denoising results of the real-world datasets. Since there is not an explicit noise magnitude $\sigma$ in the Jacobi process, we did not apply the SURE-based method (Metzler et al., 2020) to this task.

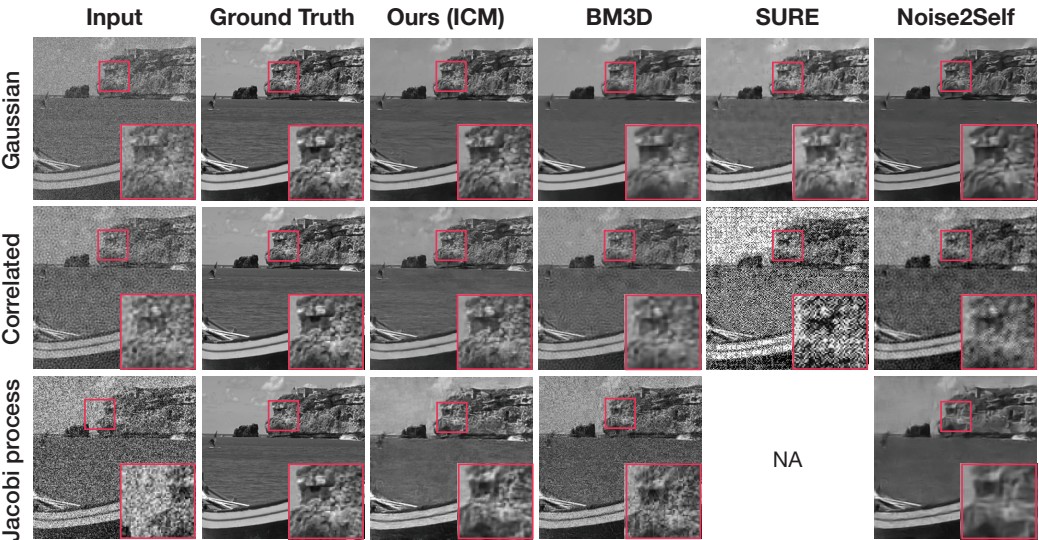

Figure 5: Denoising results of BSDS500 for natural images corrupted with three types of noise distributions. Methods compared are BM3D, SURE loss, Noise2Self, and ICM.

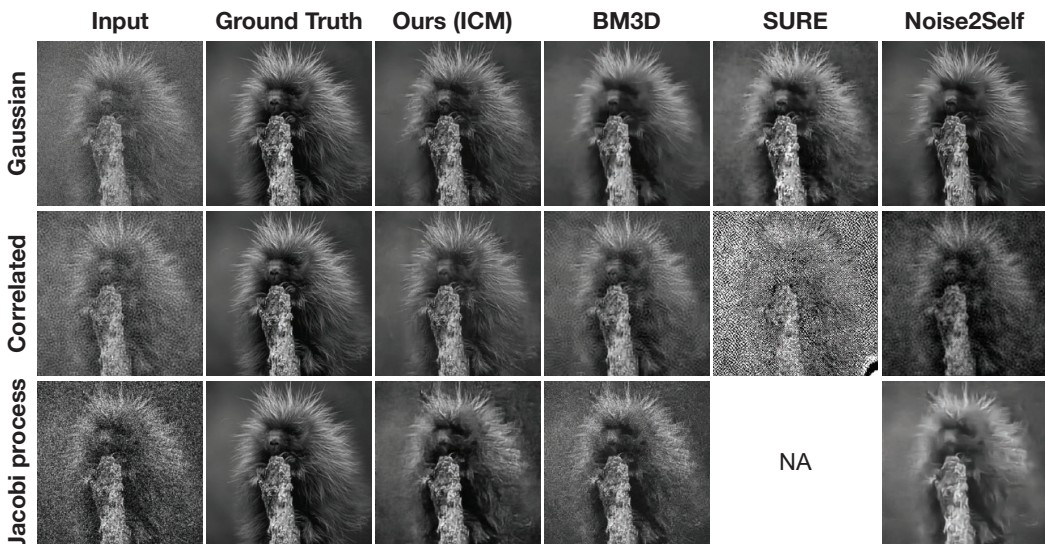

Figure 6: Denoising results of BSDS500 for natural images corrupted with three types of noise distributions. Methods compared are BM3D, SURE loss, Noise2Self, and ICM.

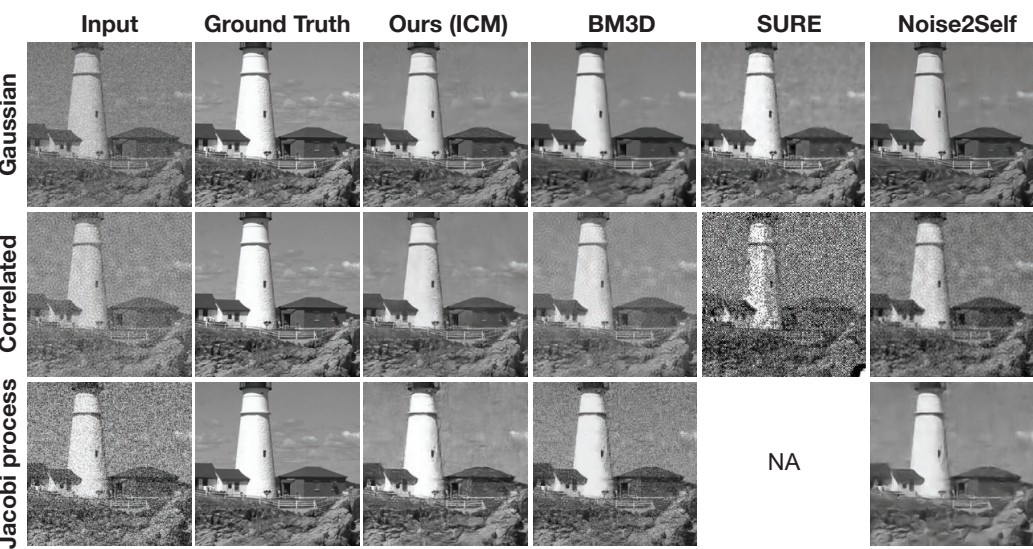

Figure 7: Denoising results of Kodak for natural images corrupted with three types of noise distributions. Methods compared are BM3D, SURE loss, Noise2Self, and ICM.

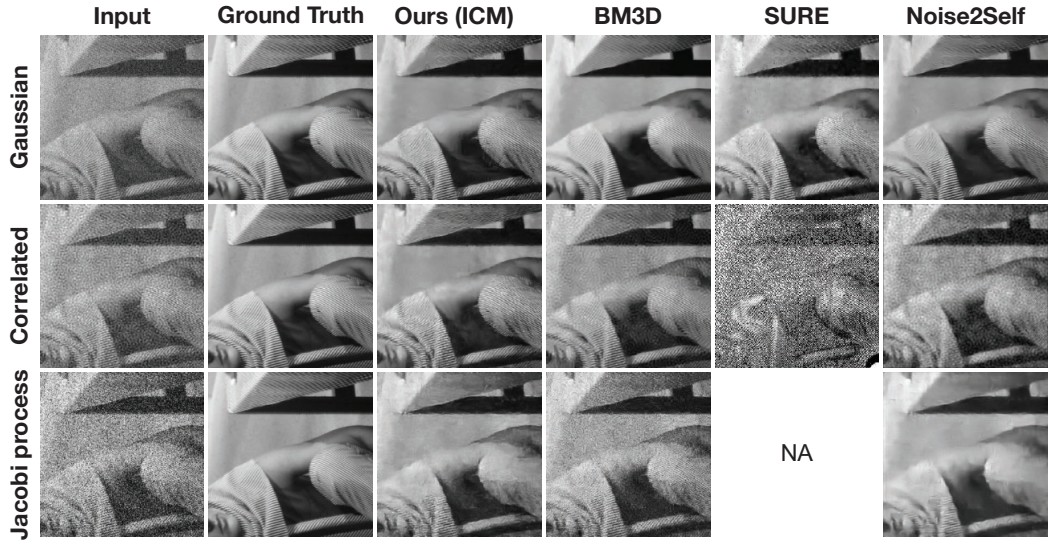

Figure 8: Denoising results of Set12 for natural images corrupted with three types of noise distributions. Methods compared are BM3D, SURE loss, Noise2Self, and ICM.

