# OpenReview forum: "Inverse Flow and Consistency Models"
_ICLR.cc/2025/Conference — Submitted to ICLR 2025_

### Official Review · Reviewer_C4GH · 2024-11-04

**Soundness:** 3
**Presentation:** 3
**Contribution:** 3
**Rating:** 6
**Confidence:** 4

**Summary:**

This paper presents a new framework, Inverse Flow (IF), designed to extend generative modeling to inverse generation tasks, such as denoising data without access to clean reference data. Traditionally, generative models like diffusion and flow-based models have relied on access to clean data to learn mappings between noisy and noise-free distributions. Generating clean data directly from noisy inputs in cases where clean data is not available is an important application of relevance in multiple areas, particularly scientific discovery, and I am glad that the authors are tackling this problem.

The framework introduces two main algorithms: Inverse Flow Matching (IFM) and Inverse Consistency Model (ICM). IFM is an approach based on solving a reverse flow via an Ordinary Differential Equation (ODE) to gradually map noisy data back to a clean state, while ICM takes a more efficient approach by using a consistency function to produce clean data without requiring iterative ODE solutions.

The authors validate IFM by testing it on both synthetic datasets and real-world applications, such as fluorescence microscopy and single-cell genomics, where it demonstrates improved performance over traditional denoising methods that require ground truth. Authors show in all examples that their IFM does better than the baselines.

**Strengths:**

+ **Originality**: The paper introduces a fresh perspective on handling inverse generation problems by leveraging continuous-time generative models without the need for clean data, which addresses a significant gap in denoising research.

+ **Quality, Clarity and Presentation**: The paper is thorough in presenting both theoretical and empirical support for its claims. Proofs for the effectiveness of IFM and ICM are detailed, and experiments are extensive, covering synthetic benchmarks and practical applications. The methods, particularly the differences between IFM and ICM, are presented clearly, with diagrams and mathematical derivations that aid understanding. The paper is accessible to readers familiar with generative models and denoising literature. That said, I do believe that the paper can benefit from a bit more explanation. I found myself re-reading Section 2.1.1, 2.1.2 and 3 a couple of times until I understood all details - explaining what was happening with an example would reduce the burden on readers.

+ **Significance**: The flexibility of IFM and ICM for a wide range of noise distributions and its applicability in denoising tasks without ground truth data offer potential advancements in fields requiring clean data from noisy measurements, such as biomedical imaging and genomics as demonstrated. Overall, the improvements in denoising performance achieved is significant, and I believe that method would foster more research in unsupervised denoising.

**Weaknesses:**

+ **Assumption on Noise Distribution**: As also noted by the authors, the methods reliance on prior knowledge of the noise distribution is a limitation, as it may restrict the method’s generalizability in scenarios where noise characteristics are unknown or complex. Given this, I would really like to understand to what extend the noise distribution should be known for the method to work well. Authors could do experiments using approximate noise distributions and see the implications on performance.

+ **How much data is required?** We know that diffusion models are generally data hungry, and particularly in scientific discovery, the amount of data available is very limited. For eg. in a typical [electron microscopy denoising](https://arxiv.org/abs/2010.12970) problem, there are only 40 frames of real data available. How does the method work when the amount of noisy data is limited?

+ **Inadequate comparison**. Can you compare with [Noise2Same](https://arxiv.org/abs/2010.11971) and [Neighbor2Neighbor](https://arxiv.org/abs/2101.02824)which doesn't make assumption on J-invariance, or justify why this comparison is not needed? This would also serve as good baselines for the experiment suggested above.

+ **The Navier-Stokes example seem a bit oversold**. In lines 317-319 the authors say that this is particularly helpful in understanding turbulent flows but the example in Fig 2 seems a bit too simple without any turbulence. The example would be much more compelling if authors apply it to an actual turbulent flow and the algorithm is able to recover the GT. Authors may find relevant dataset [here](https://turbulence.pha.jhu.edu)

**Questions:**

+ **Typo:** In equation 42, the derivate has V instead of $v$

+ **Comparative Baselines**:  Can authors include a supervised baseline in the examples to see what is performance gap with a supervised system?

+ **Is ICM training stable?** Since ICM relies on enforcing self-consistency across noise levels, it may be prone to issues with training stability or convergence in some noise settings. Clarification on the strategies used to stabilize the model (e.g., gradient clipping, regularization methods) could give further insight into ensuring training robustness.

---

> ### Author Response · Authors · 2024-11-25
>
> We thank the reviewer for constructive suggestions and have provided detailed clarifications and responses to all the identified weaknesses and questions, as outlined below.
>
> > Assumption on Noise Distribution
>
> We have now demonstrated that our method is robust to using approximate noise distribution. We added experiments on data with $\sigma=25$ Gaussian noise, but training and denoising with different noise level assumptions from $\sigma=12.5$ to $\sigma=50$. Our method demonstrates stable performance within the range of $\sigma=25$ to $\sigma=35$, indicating that overestimating the noise level has minimal impact on the model's effectiveness.
>
> **Table R5: Performance for different noise level assumptions**
> |         | sigma=12\.5 | sigma=15 | sigma=20 | **sigma=25** | sigma=30 | sigma=35 | sigma=50 |
> | :------ | :---------- | :------- | :------- | :------- | :------- | :------- | :------- |
> | BSDS500 | 21\.59      | 22\.43   |  24\.78  | 28\.16   | 28\.09   | 27\.55   | 25\.71   |
> | Kodak   | 21\.62      | 22\.49   |   25\.03  | 29\.08   | 28\.99   | 28\.43   | 26\.66   |
> | Set12   | 21\.67      | 22\.56   |   25\.14  | 29\.19   | 29\.20   | 28\.65   | 26\.86   |
>
> > How much data is required?
>
> We have added an evaluation of our method on the electron microscopy denoising dataset[1] to demonstrate its performance on small sample-size datasets. Since the original authors did not release the real experimental data, we used the simulated dataset they provided and added Poisson noise, which is the noise distribution in the real data according to their analysis. The dataset consists of 46 samples. The results indicate that our method is applicable to small datasets and outperforms other approaches in this scenario. While diffusion model is known as being data hungry, we believe that our method is efficient on sample size because it does not involve training a full generative model.
>
> **Table R6: Performance on electron microscopy denoising dataset**
> |      | Input | Noise2Void | Noise2Self | Ours (ICM) |
> |:----:|:-----:|:----------:|:----------:|:----------:|
> | PSNR | 23.70 |    38.67   |    41.42   |    43.78   |
>
> > Comparative Baselines
>
> We conducted additional experiments comparing our method with a fully supervised model with the same model architecture and training settings. The results demonstrate that our method achieves performance on par with the fully supervised model. Furthermore, we observed overfitting during the training process of the fully supervised model, a problem that did not occur with our method, highlighting the superiority of our approach. We will merge this result into Table 1 of our manuscript.
>
> **Table R7: More performance comparison**
> |                      |         | Supervised | Nbr2Nbr | Ours (ICM) |
> |:--------------------:|:-------:|:----------:|---------|:----------:|
> |                      | BSDS500 |    28.00   |  27.95  |    28.16   |
> |       Gaussian       |  Kodak  |    28.91   |  28.69  |    29.08   |
> |                      |  Set12  |    28.99   |  28.81  |    29.19   |
> |                      | BSDS500 |    27.10   |    -    |    27.64   |
> |      Correlated      |  Kodak  |    27.97   |    -    |    28.53   |
> |                      |  Set12  |    27.88   |    -    |    28.46   |
> |                      | BSDS500 |    24.34   |    -    |    24.28   |
> | SDE (Jacobi process) |  Kodak  |    25.34   |    -    |    25.07   |
> |                      |  Set12  |    25.01   |    -    |    24.74   |
>
> > Is ICM training stable?
>
> We didn’t encounter stability issues during ICM training. We utilized the AdamW optimizer with a weight decay of 0.01 to stabilize the optimization process. Additionally, we adopted a two-stage training strategy: in the first stage, the learning rate was set to 1e-4, and in the second stage, it was reduced to1e-5. Details of the training process for each dataset are provided in the supplementary materials.
>
> Reference:
>
> [1] Mohan, Sreyas, et al. "Deep denoising for scientific discovery: A case study in electron microscopy." IEEE Transactions on Computational Imaging 8 (2022): 585-597.

---

> ### Author Response · Authors · 2024-11-25
>
> > Inadequate comparison
>
> We have supplemented the comparison results with Neighbor2Neighbor in Table R7. Neighbor2Neighbor still requires pixel-wise independence and is therefore unable to handle correlated noise, leading to training collapse. Similarly, training also failed when applying Neighbor2Neighbor to the Jacobi process distribution. Although the Jacobi process is pixel-wise independent, Neighbor2Neighbor remains ineffective in this case. A similar issue was observed when handling Rayleigh noise[1]. These results demonstrate that our method outperforms Neighbor2Neighbor. The comparison experiments with Noise2Same are still in progress. We will update the results as soon as the experiments are finished. This section will be incorporated into the manuscript.
>
> > The Navier-Stokes example seem a bit oversold
>
> We are conducting experiments on the turbulent flow dataset. We will update the results later.
>
> We also thank the reviewers for pointing out the typo and we have corrected that.
>
> Reference:
>
> [1] Xie, Yutong, et al. "Unsupervised image denoising with score function." Advances in Neural Information Processing Systems 36 (2024).

---

### Official Review · Reviewer_dg2J · 2024-11-08

**Soundness:** 3
**Presentation:** 2
**Contribution:** 3
**Rating:** 3
**Confidence:** 5

**Summary:**

This paper is on the topic of learning generative models from corrupted data. The degradation operator is known but not invertible. To tackle this problem, the authors propose a new framework, called Inverse Flow Matching (IFM), and its computationally tractable version Inverse Consistency Models (ICM). In their training objective, the authors replace the (non-available) clean data with the ODE solution (simulated by a neural network) that arises when the deterministic sampling is initialized with the available corrupted samples. The authors test their method in synthetic, semi-synthetic and real datasets and show that the proposed framework outperforms Noise2X-type baselines.

**Strengths:**

1) The problem of learning generative models from corrupted data is very interesting and it has broad scientific applications.
2) The proposed framework is very general: it can handle arbitrary corruption as long as the distribution is identifiable from the measurements and the corruption model is known.
3) The proposed algorithm borrows ideas from the consistency models literature and extends them in a novel way to learn without access to uncorrupted samples.
4) The proposed method clearly outperforms the considered baselines.

**Weaknesses:**

While I strongly believe that this paper has some very interesting aspects, I think there are also several important weaknesses.

1) I believe the presentation of the paper should be improved. In the background Section 2.1.1. there are several undefined symbols. For example, $u_t(x)$ is defined in (1) as the vector field of the ODE. However, in (2) a conditional form of this vector field is used, without defining what this conditional form means and how it relates with the vector field in (1). Similarly, there is a vague reference to the $u_t(x | x_0, x_1)$ which is also not defined.

   There are also several typos throughout the paper, for example, there is a parenthesis missing in line 103.

   I also do not understand the point of lines 178-179. I would think that if you have managed to learn the prior $p_0(x_0)$ and the corruption model is known, then posterior sampling can be performed, so I do not understand why this is a strength of this particular method. The proof of Theorem 1 seems also unnecessarily convoluted. If I understand correctly the steps are: 1) show that minimizing the loss guarantees that $p_1=q_1$, 2) write the marginal expression for $q_1$, and 3) use the assumption to show that 1), 2) imply $p_0=q_0$. I think that step 1 needs more clarity. I don't understand from what's written how this follows from the minimization of the objective. Steps 2) and 3) are more straightforward and I believe that the paper would benefit by presenting them as such. Finally, I believe that it is very hard for someone not familiar with FMD and Single Cell Genomics Data to understand what's happening in section 4.3. The authors should include more details so that the authors/reviewers can appreciate the significance of the contribution, especially for Section 4.3.2.

2) The position related to prior work is weak. The authors mention that Ambient Diffusion and G-SURE-diffusion are only useful for generation and not for denoising. I respectfully disagree with this statement. In fact, there are results in the papers for solving inverse problems. The authors also miss prior work, including 1) the [Ambient Diffusion Posterior Sampling](https://arxiv.org/abs/2403.08728) paper, 2) the [Consistent Diffusion Meets Tweedie](https://arxiv.org/abs/2404.10177) paper, 3) works on learning diffusion models from corrupted data based on EM, e.g. [this work](https://arxiv.org/abs/2405.13712). Particularly 2) seems connected to the IFM objective. Overall, there are no comparisons at all with existing methods for training generative models from corrupted data. I understand the point of the authors that their technique is more general and can handle arbitrary corruptions but still, it would be very beneficial for the work to show how the method performs for corruptions that prior work can handle (such as additive Gaussian noise and masking). The authors only compare with restoration baselines (variants of the Noise2Noise framework) and they do not compare with any other generative modeling approach.

3) The experimental evaluation is limited. I believe comparisons with other generative modeling-based approaches for learning with corrupted data should be included, in the setting tested by prior work. For the experiments with natural images, the authors could have included more extensive baselines (colorized images in standard datasets such as CIFAR-10, CelebA, FFHQ, ImageNet). I also believe that the authors should spend more time explaining in the paper how Noise2Self and Noise2Void work since these are the main baselines. Finally, I firmly believe that the authors should also provide generation results (e.g. FID scores) for their trained models to understand the capacity of the method for training generative models from corrupted data.


---

**Update**: During the authors-reviewers discussion period, the authors confirmed that the trivial solution of always predicting 0 minimizes their objective. Even though this does not happen in practice, the theoretical validity of the proposed method is severely weakened. This also raises very important concerns regarding the correctness of the proposed theorems. For these reasons, I am lowering my score to 3 and I am recommending reject.

**Questions:**

See also the weaknesses above.

1) Could the authors provide connections and missing comparisons with prior work in the space of learning generative models from corrupted data?

2) Could the authors provide unconditional generation results for their method?

3) Could the authors demonstrate that their generalized consistency is useful beyond the setting of learning from corrupted data? Is this currently ablated somewhere?

4) Could the authors include experimental results in widely used datasets such as CIFAR-10, FFHQ, ImageNet?

---

> ### Author Response · Authors · 2024-11-25
>
> We appreciate the reviewer's constructive feedback and have thoroughly addressed all identified weaknesses and questions, as outlined in detail below.
>
> > About line 178-179
>
> The reason to emphasize this is that the ODE in inverse flow serves the dual function of both generating $p_0(x)$ and directly mapping $x_1$ to $x_0$ such as for denoising.  This is different from posterior sampling methods. Indeed posterior sampling is another way to estimate $x_0$ when $ p_0(x)$ is learned, but it generally requires approximation to apply. Inverse flow also does not necessitate learning a full generative model of $ x_0$, making the training more efficient.
>
> > The proof of Theorem 1
>
> Your understanding is correct. We will try to make the proof more straightforward. Concerning your question, the general idea is this: when the object is minimized, there is a continuous flow between q0 and p1 and the same flow also connects q0 and q1 since the flow is learned by the same neural network. By the one-to-one mapping of continuous flow, which maps q0 to p1 and also maps q0 to q1, we have p1 equals q1.
>
> > More background information for FMD and Single Cell Genomics Data and explain in the paper how Noise2Self and Noise2Void work
>
> We will provide more background and details about fluorescent microscopy denoising and single-cell genomics denoising as well as Noise2Self and Noise2Void in the revised manuscript.
>
> >The position related to prior work is weak…they do not compare with any other generative modeling approach.
>
> We have revised our manuscript. As the reviewer pointed out, Ambient Diffusion, G-SURE, and  the more recent Consistent Diffusion Meets Tweedie method can indeed be applied to denoising, specifically for linearly corrupted data. Ambient Diffusion Posterior Sampling only requires training on linearly corrupted data and can be applied to denoising data with other noise types. Expectation Maximization based method also allows diffusion models to learn from noisy data. Compared to these existing works, our method is more generally applicable and supports any continuous noise types, including when the available data is corrupted by other types of noises.
>
> Moreover, we have added direct comparison with a generative modeling from corrupted data method. On the subsampled 32*32 CelebA dataset with Gaussian noise ($\sigma=0.03$) similar to [1],  our method obtained a PSNR of 38.86 and G-SURE-diffusion obtained a PSNR of 36.40, which shows our method outperforms than G-SURE-diffusion in denoising. The G-SURE model was retrained on this dataset following the methodology described in its original paper.
>
> **Table R3: Denoising performance compared with generative models**
> |      | G-SURE | ICM   |
> |------|--------|-------|
> | PSNR | 36.40  | 38.86 |
>
> >The experimental evaluation is limited. I believe comparisons with other generative modeling-based approaches for learning with corrupted data should be included, in the setting tested by prior work.
>
> We have now demonstrated that our method can be extended to generative modeling with corrupted data. The extension is straightforward by extending the time schedule beyond the observed data (t=1) to when it reaches stationary/prior distribution. Specifically, in the original ICM training,  with the consistency function maps noisy data (t=1) to unobserved ground truth data (t=0) for denoising results. We extended the ICM training process to t=4, where the input approximates fully Gaussian noise, and the consistency function maps Gaussian noise to t=0 for generation results.
>
> We trained and compared our method against G-SURE-diffusion on generative modeling of the subsampled 32*32 CelebA dataset with Gaussian noise ($\sigma=0.03$) dataset adapted from [1].
>
> For generative sampling from ICM, we adopted the sampling method described in the original consistency models paper [2] and reported the FID with different NFE values. Both our method and G-SURE-diffusion were evaluated on the CelebA test set, where 10,000 samples were generated for the evaluation.
>
> **Table R4: Generative performance compared with generative models**
> |     | ICM    | ICM    | ICM    | G-SURE |
> | :-- | :----- | :----- | :----- | :----- |
> | NFE | 1      | 2      | 3      | 20     |
> | FID | 34\.90 | 25\.64 | 17\.28 |  33\.09  |
>
> Reference:
>
> [1] Kawar, Bahjat, et al. "Gsure-based diffusion model training with corrupted data." arXiv preprint arXiv:2305.13128 (2023).
>
> [2] Song, Yang, et al. "Consistency models." arXiv preprint arXiv:2303.01469 (2023).

---

> > ### Comment · Reviewer_dg2J · 2024-11-25
> > **Additional Question**
> >
> > Thank you for your response. I have an additional, but very fundamental question regarding the algorithm and the theory.
> >
> > In Algorithm 2, what prevents the network c_theta from learning a very trivial map, e.g. mapping everything to 0?
> > It looks like the objective is going to be minimized by such a trivial map.
> >
> > You mention something along the lines of "[...] the one-to-one mapping of continuous flow". How do you enforce that the network will actually learn an one-to-one mapping and not a trivial function like the one that maps everything to 0?

---

> > > ### Author Response · Authors · 2024-11-25
> > >
> > > Thank you very much for your response.
> > >
> > > ICM inherits the design of consistency models[1] whose boundary condition $c_{\theta}(x,0)=x$ precludes the trivial solution $c_{\theta}(x,t)=0$. Additionally, the residual connection in the consistency models ensures the similarity between the predicted output and the input, which aligns with the requirements of the denoising task, further preventing the model from degenerating into an a all-zeros mapping.
> > >
> > > Reference:
> > >
> > > [1] Song, Yang, et al. "Consistency models." arXiv preprint arXiv:2303.01469 (2023).

---

> ### Author Response · Authors · 2024-11-25
>
> For questions
>
> > Could the authors provide connections and missing comparisons with prior work in the space of learning generative models from corrupted data?
>
> We have now addressed this point (see reply to “The position related to prior work is weak”), including comparing with prior work in learning generative model from corrupted data (Table R3).
>
> > Could the authors provide unconditional generation results for their method? Could the authors include experimental results in widely used datasets such as CIFAR-10, FFHQ, ImageNet?
>
> We have now addressed this point (see reply to “The experimental evaluation is limited”). We provided the quantitative unconditional generation results for CelebA (Table R4) and qualitative results in the appendix of the revised manuscript.
>
> > Could the authors demonstrate that their generalized consistency is useful beyond the setting of learning from corrupted data? Is this currently ablated somewhere?
>
> Generalized consistency training enables us to extend the application of consistency models to any forward diffusion processes or conditional ODE including those that introduce non-Gaussian noise. For example, in the Dirichlet Diffusion Score Model [1], the forward diffusion process is a multivariate Jacobi process which transforms one-hot encoding of discrete data (e.g., DNA sequences) into Dirichlet stationary distribution. Such diffusion processes are not supported by the original consistency training approach but are feasible with generalized consistency training. While further practical applications of generalized consistency training are beyond the scope of this manuscript, we provide a brief illustrative example and leave further applications for future work.
>
> For the ablation question, most of our applications involved non-Gaussian noise, thus we cannot ablate GCT because it is indispensable for supporting these noise types.
>
> We believe that the additional experiments and clarifications have adequately addressed the reviewer's comments. We would appreciate any adjustment of the scores if the response is deemed satisfactory.
>
> Reference:
>
> [1] Avdeyev, Pavel, et al. "Dirichlet diffusion score model for biological sequence generation." International Conference on Machine Learning. PMLR, 2023.

---

> > ### Comment · Reviewer_dg2J · 2024-11-25
> > **Additional clarification needed**
> >
> > Could you please also reply to the additional question I asked during the rebuttal?
> >
> > > In Algorithm 2, what prevents the network c_theta from learning a very trivial map, e.g. mapping everything to 0? It looks like the objective is going to be minimized by such a trivial map. You mention something along the lines of "[...] the one-to-one mapping of continuous flow". How do you enforce that the network will actually learn an one-to-one mapping and not a trivial function like the one that maps everything to 0?

---

> ### Comment · Reviewer_dg2J · 2024-11-25
> **Still unclear**
>
> This still doesn't make sense to me. For normal consistency training, this condition should be enough because x in that case is sampled from p_0. In your case, you do not have access to p_0 so what x is sampled from depends on the previous outputs of your model. But if your model always outputs 0, then x itself will be 0, and then this condition is also satisfied.

---

> > ### Author Response · Authors · 2024-11-25
> >
> > You are correct that $p_0(x)$ being a point mass and $c$ being always equal to that constant is indeed a solution to this objective, and learning the ground truth distribution is also a solution. However, during the training process, we have never encountered any collapse, and the trivial solution does not constitute a practical barrier to the application of ICM.
> >
> > We will add a discussion about the trivial solution in the appendix. Even though it has not been a practical concern, we can enforce a regularity condition for $p_0(x)$ to prevent the trivial solution, by adding a penalty on low entropy solutions of $p_0(x)$.
> >
> > Our hypothesis is that even without the penalty, the learning dynamics favor the ground truth solution and drive the model away from the trivial solution which involves singularity. A more rigorous analysis of the model optimization process is needed to obtain a deeper understanding.

---

> > > ### Comment · Reviewer_dg2J · 2024-11-26
> > > **Lowering my score**
> > >
> > > Dear authors,
> > > since the trivial solution of always predicting 0 is a solution to your objective, the method is not theoretically valid. This also raises very important concerns about the correctness of your Theorems that claim that the underlying distribution is recovered.
> > >
> > > Because of this reason and the other weaknesses I mentioned during my review, I am lowering my score to 3.

---

> > > > ### Author Response · Authors · 2024-11-26
> > > > **The trivial solution does not invalidate the proof and the method**
> > > >
> > > > I hope we can clarify that the trivial solution does not invalidate the proof and the method, for the following reason:
> > > >
> > > > Our proof relies on the one-on-one mapping properties of ODE. The one-on-one mapping property does not hold only if the distribution collapses to a singular point. However, this only happens when $p_0(x)$ is a point mass.  Our proof remains valid, with the additional condition that the $p_0(x)$ is not a point mass, which is easy to verify and thus does not affect the validity of our method.
> > > >
> > > > The nuisance of singularity related to ODE  is common and not unique to our proof. Flow matching relies on the similar time-reversal property of ODE and this property also does not hold when the conditional ODE  $u_t(x_t|x_0)$ starts from a point mass at time 0. This issue does not invalidate the application of flow matching in practice.
> > > >
> > > > We hope that this clarifies this point. Please let us know if you think our explanation is invalid. If there are remaining weaknesses that we haven't addressed, we hope that the reviewer will point that out and we are committed to addressing those points.

---

> > > > > ### Comment · Reviewer_dg2J · 2024-11-26
> > > > >
> > > > > Dear authors,
> > > > > if I understand correctly, the difference between your work and normal flow matching is that in normal flow matching the invertibility of the ODE arises from the distribution p_0(x) itself. In your case, this distribution is actually your network's distribution, and hence even if the measurements-generating process would not have such a singularity, theory-wise, your minimizer can get stuck in this singularity point.
> > > > >
> > > > > I further do not see why this only happens when p_0(x) is a point mass. Your proof relies on the fact that the **learned** ODE is an  one to one mapping. This property is not enforced and there can be many functions that are not one to one mappings, beyond the functions that generate a point mass distribution.

---

> > > > > > ### Author Response · Authors · 2024-11-26
> > > > > >
> > > > > > We would like to clarify our method on the following two points in response to the reviewer's feedback.
> > > > > >
> > > > > > > even if the measurements-generating process would not have such a singularity, theory-wise, your minimizer can get stuck in this singularity point
> > > > > >
> > > > > > As long as the minimizer does not get trapped in a singularity, which is straightforward to verify in practice and has been consistently validated in our extensive experiments, the result should recover the ground truth distribution.
> > > > > >
> > > > > > > ... learned ODE is an one to one mapping. This property is not enforced ...
> > > > > >
> > > > > > Any learned ODE without singularity does indeed guarantee one-to-one mapping. This result is the basis of neural ODE[1].
> > > > > >
> > > > > > Reference:
> > > > > >
> > > > > > [1] Kidger, Patrick. "On neural differential equations." arXiv preprint arXiv:2202.02435 (2022).

---

> ### Author Response · Authors · 2024-11-25
>
> We are currently revising the manuscript. You can view the generative results of our model via the following URL https://www.dropbox.com/scl/fi/3bjtb2m99d7dwa4mq1b5o/CT_best_max10min_10k.pdf?rlkey=v1dkj511z72utxb4jg7bjk0sx&st=mp0kaqow&dl=0 (created by an anonymous Dropbox account)

---

> ### Author Response · Authors · 2024-12-03
> **Response Summary and Updates in Revised Manuscript**
>
> As the discussion period is nearing its end, we kindly remind the reviewer we have further responded to your concerns in the general comment. In the revised manuscript, we added more discussion about prior work in section 2.1.3. We also added Remark 1 after the proof in section A.2.1 to address the problem of the trivial solution.

---

### Official Review · Reviewer_TD3q · 2024-11-10

**Soundness:** 2
**Presentation:** 3
**Contribution:** 1
**Rating:** 6
**Confidence:** 3

**Summary:**

This work considers the problem of denoising noisy observations $x_1$ without access to ground truth observations $x_0$. This work proposes two methods to solve this problem:
1. Inverse Flow Matching (IFM) training: The inverse flow matching loss is similar to the regular flow matching loss with two key differences: As $x_0$ is not accessible, ODE is simulated from $x_1$ to $x_0$ to approximate $x_0$. The conditional vector field used is the standard conditional optimal transport flow.
2. Inverse Consistency Model (ICM) training: This is the inverse counterpart of the standard consistency distillation. The model is trained to predict $x_0$ given $x_t$ where $t \in [0, 1]$.  This predicted $x_0$ is used to predict $x_{t+1}$ and $x_{t}$ (by applying one step of Euler ODE solver). The loss between
Both the proposed methods have been tested on a combination of standard image datasets such as BSDS500, Kodak, Set12, scientific datasets like microscopy dataset and genomics data, as well as synthetic datasets such as Navier-Stokes, gaussian datasets.

**Strengths:**

- Inverse consistency model (ICM) training is simulation-free. It also gives good qualitative results in gray scale as shown in the appendix, and quantitative results as shown in Table 1 compared to prior baselines.
- The proposed method works better compared to prior methods on complex noise such as correlated noise and on data transformed with Jacobi process.
- The proposed methods are easy to understand as these are simple modifications of existing methods.

**Weaknesses:**

- Inverse Flow Matching training requires simulating the ODE which can be slow.
- The proposed methods are simple modifications of existing methods such as consistency distillation/training and flow matching and there seems to be limited novelty.
    - Generalized Consistency Training: The core idea is same as consistency distillation. This work rewrites consistency distillation loss with a general ODE instead of Probability Flow ODE (PF-ODE).  Instead of assuming a pretrained diffusion model, GCT assumes that the vector field is user-specified (which is equivalent to having a pretrained teacher model).

- More experiments are needed, especially to measure robustness of the proposed method.
    - Different levels of noise (for eg. in case of Gaussian noise, consider different values of variance - both low and high)
    - Different combinations of noise - What happens if we have multiple types of noise being applied simultaneously in the observed image? Prior works also consider more types of noise: Noise2Score (Xie et al. 2023) presents results on both additive gaussian noise and multiplicative noise such as gamma noise, poisson noise and rayleigh noise. Please refer to Table 1 in this paper for a more extensive list of all different types of noise the authors consider (around 16 types).
- More experiments on colored datasets. Prior works consider results on extensively colored datasets such as ImageNet 256X256 (used by Noise2Same), CBSD68 dataset (Noise2Score). However, this work only shows qualitative results on fluorescence microscopy images which doesn’t have lot of color. Kodak and BSDS500 are all colored but qualitative results shown are in gray scale (Figure 5,6,7).
- More experimental details are needed. For instance, the paper should mention the architecture used to train these models on each of the datasets. Various hyperparameter values for training on real-world datasets should be included,

**Questions:**

1. Kodak and BSDS500 datasets are colored - why weren't colored images included in qualitative results? Were the models trained on grayscale images?
2. I think Noise2Score (Kim & Ye, 2021) should be included as a baseline. It seems to outperform other baselines considered in this work; though it is unclear if it will perform well with the types of noise considered in this work.

---

> ### Author Response · Authors · 2024-11-25
>
> We thank the reviewer for the thoughtful comments. We have now clarified and responded to all weaknesses and questions by the reviewer, as detailed below.
>
> > Inverse Flow Matching training requires simulating the ODE which can be slow.
>
> We have addressed the training speed problem of IFM by proposing ICM. For IFM, one epoch takes 5.2s on average and for ICM one epoch takes 2.2s. The performances by PSNR are similar (shown below) with ICM 2x faster during training and 10x faster during inference. ICM's speed advantage will be higher if more steps are required in the ODE solver (10 steps were used due to computational cost). The user can choose the best method for their problem based on their model size and computational budget.
>
> |         |  Input  |  ICM  |  IFM  |
> |:-------:|:-----:|:-----:|:-----:|
> | BSDS500 | 20.17 |28.16 | 28.33 |
> |  Kodak  | 20.18 |29.08 | 29.25 |
> |  Set12  | 20.16 |29.19 | 29.34 |
>
> > The proposed methods are simple modifications of existing methods such as consistency distillation/training and flow matching and there seems to be limited novelty.
>
> The simplicity and generality of our method is by design. This allows applying a wide range of diffusion, flow matching, and consistency model methods to the inverse flow problem.
>
> We addressed an unsolved problem of providing a general solution to the inverse flow problem: It was not obvious that flow matching or diffusion can recover data from the earlier timepoint without requiring data for that timepoint [1] and we showed to our knowledge the first general solution to this problem applicable to the entire family of continuous time generative models [2][3][4]. Our discovery of the effectiveness of inverse flow matching and inverse consistency models opens the door for new applications.
>
> For Generalized Consistency Training, we would like to clarify that the biggest difference between GCT and consistency distillation is that GCT only requires a conditional ODE $u_t(x_t \mid x_0)$ or $u_t(x_t \mid x_0, x_1)$  which can be easily specified by user without any training (similar to conditional flow matching), rather than an unconditional ODE $u_t(x_t)$ which has to be obtained from pretraininig.Therefore the applications are different since GCT can be applied without any pre-training.
>
> We will clarify these points in the revised manuscript.
>
> Reference:
>
> [1] Chung, Hyungjin, et al. "Diffusion posterior sampling for general noisy inverse problems." arXiv preprint arXiv:2209.14687 (2022).
>
> [2] Lipman, Yaron, et al. "Flow matching for generative modeling." arXiv preprint arXiv:2210.02747 (2022).
>
> [3] Ho, Jonathan, Ajay Jain, and Pieter Abbeel. "Denoising diffusion probabilistic models." Advances in neural information processing systems 33 (2020): 6840-6851.
>
> [4] Song, Yang, et al. "Consistency models." arXiv preprint arXiv:2303.01469 (2023).

---

> ### Author Response · Authors · 2024-11-25
>
> > More experiments are needed, especially to measure robustness of the proposed method. More experiments on colored datasets.
>
> We have now added extensive experiments to measure how different levels of Gaussian noise, different combinations of noises, affect performance. Our method is robust over all noise levels and noise type combinations we applied.
>
> **Table R1: Denoising performance for different levels of Gaussian noise measured by PSNR in dB**
> |         | $\sigma=5$ Input | $\sigma=5$ Prediction | $\sigma=12.5$ Input | $\sigma=12.5$ Prediction | $\sigma=25$ Input | $\sigma=25$ Prediction | $\sigma=50$ Input | $\sigma=50$ Prediction | $\sigma=75$ Input | $\sigma=75$ Prediction |
> |:-------:|:----------------:|:---------------------:|:-------------------:|:------------------------:|:-----------------:|:----------------------:|:-----------------:|:----------------------:|:-----------------:|:----------------------:|
> | BSDS500 |       34.15      |         37.56         |        26.19        |           31.85          |       20.17       |          28.16         |       14.15       |          24.98         |       10.63       |          23.33         |
> |  Kodak  |       34.15      |         37.92         |        26.19        |           32.56          |       20.18       |          29.08         |       14.15       |          25.96         |       10.63       |          24.33         |
> |  Set12  |       34.15      |         37.87         |        26.20        |           32.78          |       20.16       |          29.19         |       14.13       |          25.78         |       10.63       |          23.86         |
>
> **Table R2: Denoising performance for different noise distributions measured by PSNR in dB**
> |     Noise type    |         | Input | Noise2Void | Noise2Self | Noise2Score | Ours (ICM) |
> |:-----------------:|:-------:|:-----:|:----------:|:----------:|:-----------:|:----------:|
> |      Poisson      | BSDS500 | 23.78 |    28.29   |    28.52   |  **30.53**  |    29.91   |
> |      $\zeta=0.01$  |  Kodak  | 23.60 |    28.76   |    29.36   |  **31.10**  |    30.58   |
> |                   |  Set12  | 23.08 |    30.01   |    29.23   |  **30.94**  |    30.68   |
> |       Gamma       | BSDS500 | 26.75 |    29.17   |    27.43   |    31.14    |  **32.48** |
> |        $k=100$     |  Kodak  | 26.67 |    30.26   |    28.26   |    31.67    |  **32.97** |
> |                   |  Set12  | 25.53 |    30.44   |    28.54   |    31.21    |  **33.08** |
> |      Rayleigh     | BSDS500 | 14.03 |    28.57   |    14.86   |    30.37    |  **30.55** |
> |       $\sigma=0.3$   |  Kodak  | 13.95 |    29.73   |    14.83   |    30.96    |  **31.16** |
> |                   |  Set12  | 12.81 |    29.98   |    13.74   |    30.89    |  **31.17** |
> |  Poisson+Gaussian | BSDS500 | 22.40 |    26.45   |    27.76   |    28.54    |  **29.26** |
> |                   |  Kodak  | 22.25 |    27.67   |    28.86   |    29.02    |  **30.02** |
> |                   |  Set12  | 21.88 |    27.81   |    29.23   |    29.10    |  **30.03** |
> |   Gamma+Gaussian  | BSDS500 | 24.29 |    27.98   |    26.10   |    29.34    |  **30.53** |
> |                   |  Kodak  | 24.24 |    28.99   |    27.08   |    29.90    |  **31.22** |
> |                   |  Set12  | 23.62 |    29.53   |    26.84   |    29.69    |  **31.27** |
> | Rayleigh+Gaussian | BSDS500 | 13.85 |    28.01   |    14.72   |    29.36    |  **29.79** |
> |                   |  Kodak  | 13.77 |    29.12   |    14.69   |    30.12    |  **30.49** |
> |                   |  Set12  | 12.78 |    26.81   |    13.59   |    29.82    |  **30.50** |
> |    GaussianRGB    | BSDS500 | 20.17 |    29.72   |    27.33   |    28.28    |  **29.99** |
> |      $\sigma=25$   |  Kodak  | 20.17 |    30.65   |    28.21   |    28.66    |  **30.73** |
>
> > More experimental details are needed.
>
> We will add these experiment details in the revised manuscript.
>
> For specific questions:
>
> > Kodak and BSDS500 datasets are colored - why weren't colored images included in qualitative results? Were the models trained on grayscale images?
>
> Our original models were trained on grayscale images. We now added experiment results for Gaussian noise on colored images in Table R2, which shows our method is better than compared methods.
>
> >I think Noise2Score (Kim & Ye, 2021) should be included as a baseline.
>
> We have now included Noise2Score (Kim & Ye, 2021, Xie et al. 2023) as a baseline (see Table R2) for different combinations of noise. We have an advantage over Noise2Score for most noise types. However, Noise2Score cannot be explicitly applied to the correlated Gaussian noise discussed in our manuscript, which is based on Fourier space and also the SDE-based Jacobi process.

---

> ### Author Response · Authors · 2024-11-25
>
> We believe these additional experiments and clarifications have addressed the reviewer’s comments, and we would appreciate any adjustment of the scores if you find the response satisfying.

---

> ### Comment · Reviewer_TD3q · 2024-12-01
>
> Dear authors,
>
> Thank you for answering my questions and concerns.
> - Additional results on colored datasets for Kodak, BSD500 and Set12 are encouraging.
> - Additional results on multiple levels of Gaussian noise and multiple types of noise look promising.
> - Thank you for including Noise2Score as a baseline.
>
> I am increasing my score due to the additional experiments provided by the authors which look encouraging. I hope that authors will include these results in the paper.
>
> I am unable to increase my score beyond 5 due to a valid point raised by Reviewer dg2J about the trivial mapping being an acceptable solution for IFM/ICM. I agree that Neural ODEs are diffeomorphic maps by definition. However, in this work, input x1 is a noisy image, and therefore, the assumption of diffeomorphism won't be valid for tasks considered in this work such as denoising. Further, many inverse problems by definition are ill-posed. Therefore, I am unsure about the validity and practicality of the assumptions made in theoretical work.

---

> ### Author Response · Authors · 2024-12-03
> **Response Summary and Updates in Revised Manuscript**
>
> As the discussion period is nearing its end, we kindly remind the reviewer we have further responded to your concerns in the general comment. In the revised manuscript, we added additional experiment results in the appendix A.5. We also added Remark 1 after the proof in section A.2.1 to address the problem of the trivial solution.

---

> > ### Comment · Reviewer_TD3q · 2024-12-03
> >
> > Dear authors, Thank you for additional clarification. I feel that the assumption of knowledge of the noisy distribution $p(x_1)$ is unrealistic in many practical scenarios. Further, assumption that the noisy data distribution $p(x_1)$ must be transformable from $p(x_0)$ via an ODE  is also limiting. Many noise distributions seen in practice are discrete in nature such as salt and pepper noise, poisson noise, quantization noise e.g. in jpeg compression, and a good transformation via an ODE might not exist in such cases. Interestingly, from the results above, we know that ICM doesn't work very well for poisson noise. These assumptions and limitations need to be stated explicitly.
> >
> > I do agree that many methods seen in practice such as in self-supervised learning and EM accept trivial solutions and need good initialization or other tricks to prevent learning of trivial solution. I would like to encourage authors to include a list of best practices to prevent learning of collapsed/trivial solutions. (for eg. Does use of EMA prevent collapse similar to self-supervised learning?)
> >
> > Further, certain claims need to be toned down. As Reviewer C4GH has pointed out, Navier Stokes solutions seems to be for a specific simple case of planar flow for high viscosity fluids and not for turbulent flows. I hope that the authors will make these necessary corrections.
> >
> > Finally, advantages of generalized consistency training hasn't been demonstrated to the full extent (e.g. the claim - "GCT extends consistency training to any conditional ODE or forward diffusion process" - while true theoretically hasn't been demonstrated to the full extent in practice.)

---

> > > ### Author Response · Authors · 2024-12-04
> > > **Thank you for the constructive feedback!**
> > >
> > > >I feel that the assumption of knowledge of the noisy distribution $p(x_1)$ is unrealistic in many practical scenarios. Further, assumption that the noisy data distribution $p(x_1)$ must be transformable from $p(x_0)$ via an ODE is also limiting. Many noise distributions seen in practice are discrete in nature such as salt and pepper noise, poisson noise, quantization noise e.g. in jpeg compression, and a good transformation via an ODE might not exist in such cases. Interestingly, from the results above, we know that ICM doesn't work very well for poisson noise. These assumptions and limitations need to be stated explicitly.
> > >
> > > We are happy to state these limitations and assumptions explicitly. Assumptions on knowing $p(x_1)$ can be relaxed for approximate recovery of $p(x_0)$, though developing a theory for that is beyond the scope of this manuscript. The family of transformation that can be expressed via an ODE is still very broad and importantly includes many noise types not supported by previous methods. The limitation on discrete noises can be addressed via dequantization techniques.
> > >
> > >
> > > >I do agree that many methods seen in practice such as in self-supervised learning and EM accept trivial solutions and need good initialization or other tricks to prevent learning of trivial solution. I would like to encourage authors to include a list of best practices to prevent learning of collapsed/trivial solutions. (for eg. Does use of EMA prevent collapse similar to self-supervised learning?)
> > >
> > >
> > > Thank you for acknowledging our point. We will include the best practices.
> > >
> > > >Further, certain claims need to be toned down. As Reviewer C4GH has pointed out, Navier Stokes solutions seems to be for a specific simple case of planar flow for high viscosity fluids and not for turbulent flows. I hope that the authors will make these necessary corrections.
> > >
> > > >Finally, advantages of generalized consistency training hasn't been demonstrated to the full extent (e.g. the claim - "GCT extends consistency training to any conditional ODE or forward diffusion process" - while true theoretically hasn't been demonstrated to the full extent in practice.)
> > >
> > > We agree with these points and will revise in the final version if this manuscript is accepted for publication.

---

### Author Response · Authors · 2024-12-02
**Response to the trivial solution**

Thank you for the discussion. As both Reviewer TD3q and Reviewer dg2J’s remaining concerns hinge on the trivial solution, we have combined our responses here.

Why the trivial solution does not invalidate our theory or practicality of the algorithms:

1. The trivial solution exists for ICM, not for IFM. Our theories which build upon IFM remain valid.

2. For ICM, as proven for consistency models (Theorem 1 & 2 in [1]), as long as the consistency function does not collapse to the trivial solution, learning the consistency function is equivalent to learning the underlying ODE, making ICM equivalent to IFM. Detecting collapse is straightforward and does not constitute a practical barrier. Additionally, we would like to point out that the trivial solution exists for many valid ML algorithms, such as EM. Similar to EM, with proper initialization, we have never observed the trivial solution in practice.

Additional Clarifications

- Theorem 1: Our Theorem 1 addresses recovering the ground truth distribution $p(x_0)$ from $p(x_1)$ using the learned unconditional ODE in IFM, which is a diffeomorphic map. With Lemma 1, we further show that the learned unconditional ODE can be interpreted as the expectation of the conditional ODE over $x_0$. This interpretation explains the denoising ability of our method, as supported by extensive experimental results.

- Regarding the ill-posed inverse problem: It is true that many inverse problems are ill-posed and impossible to address perfectly. The assumptions in our Theorem 1 in fact provided some guidance on the conditions under which IFM can recover the ground truth distribution. These conditions are:

    a. Knowledge of the noisy data distribution $p(x_1)$: Either directly or through access to sufficient noisy data.

    b. Transformability of $p(x_1)$: The noisy data distribution $p(x_1)$ must be transformable from $p(x_0)$ via an ODE. This condition accommodates nearly any continuous noise distribution but excludes ill-posed transformations that lack a one-on-one mapping between $p(x_0)$ and $p(x_1)$ (e.g., transformations like converting color images to grayscale).

    We will make these conditions more explicit in the final version.

We thank both reviewers for their valuable feedback, which has helped us better understand and present our work. We believe these points address both reviewers' concerns, and we are happy to respond to any remaining questions.

Reference:

[1] Song, Yang, et al. "Consistency models." arXiv preprint arXiv:2303.01469 (2023).

---

### Meta-Review · Area_Chair_VoTV · 2024-12-18

**Metareview:**

This paper suggests inverse flow matching (IFM) and inverse consistency model (ICM) as a method to learn an approximate inverse of a corruption process with a known corruption kernel p(x1|x0) (here x0 is data and x1 is noise) and a known noise distribution p(x1). Clean samples x0 are not assumed to be given and the algorithm uses x0 computed by solving the ODE with the learned vector field. The method is justified by providing a correctness theorem that builds upon an assumed uniqueness of the data distribution under the fixed corruption kernel and noise distribution. While the reviewers appreciated the core idea, **all** reviewers expressed different concerns that, while some were addressed during the rebuttal period, some are still standing. The validity of the uniqueness assumption (which includes the trivial solution issue, but is broader than that) is questionable in practice and the authors did not sufficiently investigate its practical and theoretical implications. Other concerns include the practicality of the assumptions of known corruption kernel and noise distribution, investigation of valid initialization strategies, and too strong claims on certain results (e.g., Navier-Stokes). Overall there was not sufficient support among reviewers to accept this paper at its current state, and we encourage the authors to address reviewers’ comments in a next submission of this paper.

**Additional Comments On Reviewer Discussion:**

No additional comments.

---

### Decision · Program_Chairs · 2025-01-22

Reject